# EvoPress: Accurate Dynamic Model Compression via Evolutionary Search

Oliver Sieberling [1]   Denis Kuznedelev [2]   Eldar Kurtic [3 4]   Dan Alistarh [3 4]

## Abstract

The high computational costs of large language models (LLMs) have led to a flurry of research on LLM compression, via methods such as quantization, sparsification, or structured pruning. A new frontier in this area is given by *dynamic, non-uniform* compression methods, which adjust the compression levels (e.g., sparsity) per-block or even per-layer in order to minimize accuracy loss, while guaranteeing a global compression threshold. Yet, current methods rely on estimating the "importance" of a given layer, implicitly assuming that layers contribute independently to the overall compression error. We begin from the motivating observation that this independence assumption does not generally hold for LLM compression: pruning a model further may even significantly recover performance. To address this, we propose EvoPress, a novel evolutionary framework for dynamic LLM compression. By formulating dynamic compression as a general optimization problem, EvoPress identifies optimal compression profiles in a highly efficient manner, and generalizes across diverse models and compression techniques. Via EvoPress, we achieve state-of-the-art performance for dynamic compression of Llama, Mistral, and Phi models, setting new benchmarks for structural pruning (block/layer dropping), unstructured sparsity, and quantization with dynamic bitwidths. Our code is available at https://github.com/IST-DASLab/EvoPress.

## 1. Introduction

Compression has become a standard way of reducing the deployment costs of large language models (LLMs). Current post-training techniques can be roughly categorized into quantization-based, which reduce the bit-width of weights or activations, e.g. (Frantar et al., 2022; Lin et al., 2023; Dettmers & Zettlemoyer, 2022; Tseng et al., 2024), pruning-based, which sparsify the weight matrices, e.g. (Frantar & Alistarh, 2023; Yin et al., 2024), or structured pruning / layer dropping, which drop entire model components, e.g. (Kim et al., 2024; Men et al., 2024). While improvements are still being made, existing methods are reaching diminishing returns in terms of accuracy-vs-compression (Dettmers et al., 2023; Tseng et al., 2024).

In this context, a new direction is *dynamic*, or *non-uniform*, layer-wise compression, in which different layers can be compressed to various levels, according to their "sensitivity" relative to the model output. Dynamic compression allows to maximize model accuracy while satisfying a given compression requirement, e.g. a target model size. Instance-specific solutions for this problem have already been proposed for essentially every compression type: sparsity (Yin et al., 2024), quantization (Frantar & Alistarh, 2022), or layer dropping (Kim et al., 2024; Men et al., 2024). Broadly, these approaches work by assigning an *error/sensitivity score* to each layer and compression level, which measures the impact of its compression on output loss increase. Then, one calculates a compression assignment which minimizes the sum of error scores, while still satisfying the global compression constraint. Thus, such approaches inherently assume *error monotonicity*: i.e., that a lower sum of error scores taken over layers implies a lower compression error for the entire model.

Our work starts from the observation that error monotonicity *does not hold* generally for LLM compression: specifically, there are instances where *compressed models with lower sums of per-layer errors can perform* worse *than models with higher error*. We illustrate this fact in Table 1, which shows an instance of a layer dropping configuration where pruning *more blocks* leads to significantly better perplexity than an instance which prunes *strictly fewer* blocks.

**Contribution.** This refutation of error monotonicity implies that most prior approaches, which are based on this assumption, can lead to sub-optimal solutions. Thus, it motivates our investigation of alternatives towards optimal non-uniform compression. For this, we propose a new evolutionary search approach called EvoPress, which is provably convergent, and is also sample and iteration efficient. Thus,

[1]ETH Zürich [2]Yandex Research [3]IST Austria [4]Red Hat AI. Correspondence to: Dan Alistarh <dan.alistarh@ist.ac.at>.

*Proceedings of the 42nd International Conference on Machine Learning*, Vancouver, Canada. PMLR 267, 2025. Copyright 2025 by the author(s).

*Table 1.* Depth pruning is not monotone. In this example (Llama-3-8B), removing strictly more blocks (depicted in orange) *can improve* perplexity across sources. The left half of a block corresponds to the attention layer, the right half to the MLP.

| Model | Configuration (Each block contains Attention + MLP) | Wiki2↓ | C4↓ | FW↓ |
|---|---|---|---|---|
| | ▰▰▰▰▰▰▰▰▰▰▰▰▰▰▰▰▰▰▰▰▰▰▰▰➡ | 5.54 | 8.80 | 7.72 |
| Llama-3-8B | ▰▰▰▰▰▰▮▮▰▮▰▰▰▰▰▰▮▮▮▰▰▰▰▰▰▮▮➡ | 188.01 | 147.25 | 70.46 |
| | ▰▰▰▰▰▰▮▮▮▰▰▰▰▮▮▮▮▰▰▰▰▰▮▮▰▰➡ | **24.39** | **35.53** | **26.24** |

EvoPress is the first non-uniform compression method with guarantees; its two efficiency properties are critical for practicality in the context of LLMs, where the cost of evaluating single models ("offspring") is exceedingly high. We validate the approach across all three popular approaches for post-training LLM compression: layer dropping, one-shot sparsification, and quantization. We find that EvoPress consistently improves upon existing techniques, with major improvements at higher compression ratios.

In more detail, we assume a setting where we are given a pre-trained model, a compression constraint such as the target model size, a set of compression options (e.g., 10 possible sparsity options per layer), and aim to identify a per-layer assignment which satisfies the constraint, while minimizing accuracy loss, measured in perplexity or in-context learning accuracy degradation. As is standard, e.g. (Frantar & Alistarh, 2022), from the compression options we build a *level database*, where each layer is compressed independently to each compression option. During the candidate search, our *offspring* are models stitched together from the level database, and our *fitness function* will be the difference (e.g., in KL-divergence) between the outputs of the offspring and the original model, on a set of calibration samples.

At each step, our search algorithm starts with a single search point (candidate model), and generates a constant $\lambda \geq 1$ additional offspring, by applying a mutation operation which preserves the compression constraint. The selection stage is composed of multiple steps, where we *iteratively* evaluate the offspring and parent on *increasingly many* randomly chosen samples. For instance, we may start to evaluate the parent and $\lambda = 64$ offspring *on less than a single sample* on the first sub-step, but progressively multiply the number of calibration samples as we sift through candidates, reducing variance as we obtain more competitive offspring. We found this trade-off between exploration and evaluation variance essential for efficiency on LLMs, as it drastically reduces our total number of evaluations relative to the case where all the initial offspring must be evaluated on a full batch.

Our approach builds on the observation that the partial effectiveness of prior work, which assumes error linearity, suggests that the fitness landscape induced by compression allocation has properties similar to those of a linear function. Such functions are particularly well-suited to *hill climbing*,

meaning they can be optimized through highly local exploration, akin to gradient descent in a continuous search space. EvoPress is specifically designed as an efficient hill climber, though it is capable of optimizing a broad class of fitness environments. Notably, our algorithm guarantees convergence: specifically, any linear fitness function defined on the $n$-dimensional hypercube will be maximized in expected $O(k(n - k)/\lambda)$ generations under the constraint $\|x\|_1 = k$, where $\lambda$ is the number of offspring. The proof is quite non-trivial, as it needs to adapt stochastic drift analysis techniques to the case where multiple offspring are examined in each sub-step. In Figure 1, we illustrate the algorithm's fast convergence and high efficiency on a practical example with correlated block dropping on Llama-3-8B, where we determined the optimum via (expensive) exhaustive search: EvoPress is able to reach the optimum in only 6 generations, using a total of only 56 model evaluations. A key advantage of our approach is that it is agnostic of the model architecture and compression type. We illustrate this via experiments, which are the first to span all three compression methods, across different LLM families.

Results show that EvoPress significantly improves upon all prior work on depth pruning in terms of accuracy-vs-compression, especially at medium levels, and also outperforms the prior best methods - OWL and dynamic programming, respectively - for non-uniform pruning and quantization. Moreover, it can do so efficiently: the full version of EvoPress, applied at high compression granularity, will converge in a few hours on a single RTX 3090 GPU, and we also present a lightweight version which utilizes fewer samples and converges in $\sim 1$ hour in the same setting, on an 8B-parameter model.

## 2. Related Work

To our knowledge, we are the first to present a unified approach which covers all types of post-training LLM compression (i.e., layer dropping / depth pruning and non-uniform pruning / quantization) - so far, these problems have generally been approached independently.

**Depth Pruning.** Recently, there has been a lot of interest in compression by removing entire transformer blocks, both for efficiency and to gain insights about the language model

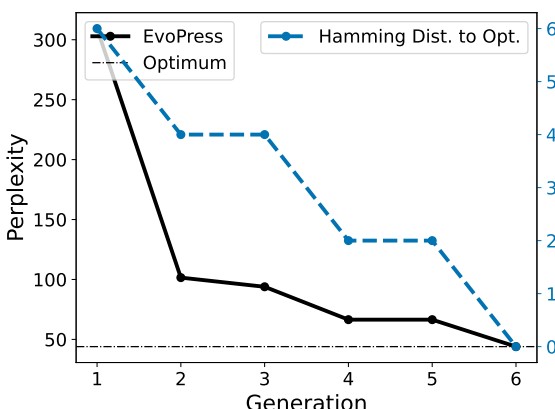

*Figure 1.* Removing twelve transformer blocks from Llama-3-8B under the constraint that only pairs of consecutive blocks can be removed. EvoPress finds the optimal configuration from the 8008 possible removal combinations in generation 6.

itself. Most methods are based on scoring the importance of each block, and then maximizing the importance of the resulting model by removing the blocks of lowest importance. Therefore, they assume error linearity, meaning that each block contributes independently to the total compression error. Weight Subcloning (Samragh et al., 2023) proposed a multi-step process to find good initializations for an untrained smaller model given an already trained larger one, where the importance of each block is scored based on the ratio of $\ell_2$ norms between the output embeddings of the block with and without the residual connection. Shortened Llama (Kim et al., 2024) proposes scoring each block by measuring the perplexity after removing the respective block from the full model. ShortGPT (Men et al., 2024) uses the cosine similarity between the input and output embeddings of each block to assess its importance. By contrast, Gromov et al. (2024) restrict to removing *consecutive blocks* and score each configuration by cosine similarity.

**Non-Uniform Pruning and Quantization.** He et al. (2018); Ashok et al. (2018) were among the first to consider automatic optimization of non-uniform compression, specifically for the case of pruning, where they developed Reinforcement Learning (RL)-based approaches. However, both approaches suffer from high tuning complexity and would be very hard to scale to large models. Follow-up work (Hubara et al., 2021; Yao et al., 2021; Li et al., 2021) considered a similar problem specifically for quantization, but explore computationally-expensive solvers (e.g. ILPs) which rely on the fact that quantization has only a small number of choices (precision levels) per layer. SPDY (Frantar & Alistarh, 2022) considered a unified framework which reduces the problems to knapsack-type instances, and solves them optimally modulo discretization. However, SPDY ex-

plicitly relies on monotonicity and linearity assumptions on the dependency between the per-layer errors and model output error, which we find not to hold on large models, especially in the high-compression regime (e.g., below 3 bits per parameter). Relative to SPDY, EvoPress provides guarantees for a broader class of input functions, and focuses on efficiency for LLM compression.

The recent OWL method (Yin et al., 2024) focuses on non-uniform pruning of LLMs, and provides consistent improvements over uniform profiles via a layer scoring system which analyzes the activation outlier structure. Experimentally, we find that OWL is effective especially for Llama-family models (Touvron et al., 2023) and at moderate sparsities, but observe significant gaps in favor of EvoPress across all models and compression levels.

**NAS and Structural Pruning.** Random search is also popular in the context of structural pruning and Neural Architecture Search (NAS) (Chen et al., 2020; Dong et al., 2021; Wang et al., 2020; Xu et al., 2021; Yin et al., 2021; Molchanov et al., 2022; Kurtić et al., 2024). However, such methods rely on re-training and have notoriously high costs, which limits their applicability to post-training compression. Thanks to its low sample complexity, we believe that EvoPress could be extensible to lightweight NAS, and plan to investigate this in future work.

## 3. Method

All applications of EvoPress are grounded in a unified framework, where the objective is to identify the optimal model that adheres to a specified compression method and constraint. Formally, given a base model $M$, we seek to maximize the performance of the compressed model while satisfying the compression constraint:

$$\hat{M}^* = \arg\max_{\hat{M}} f(\hat{M}) \quad \text{subject to} \quad g(\hat{M}) \le C,$$

where $f(\hat{M})$ quantifies the performance of the compressed model $\hat{M}$ and $g(\hat{M})$ represents the compression constraint. For simplicity, we will define $g$ as the model's total size (in terms of parameters); however, the method can be adapted to other constraints, such as inference speed.

We approach this optimization problem using evolutionary search, which is a specific form of randomized search. The feasibility of such an approach heavily depends on two factors: the time required to evaluate the fitness of a candidate solution and the number of such function evaluations needed until a satisfying result is achieved. This poses a particular challenge in our case, as assessing the performance of an LLM involves substantial computational costs.

**Level Database.** As a first step, we compress the model to different levels. It is crucial that the units we search over – specifically layers or blocks – are compressed independently; otherwise, we risk losing performance when stitching together the compressed model. Ideally, the difference between two compression levels should be consistent across layers. This uniformity simplifies the optimization process, allowing for the free exchange of compression levels, as we will demonstrate for unstructured sparsity. However, this restriction is not essential for the search procedure to be effective. In the context of quantization, we will demonstrate a relaxation of this requirement, where compression steps are uniform only across layers of the same size.

**Fitness Environment.** Given the specified database, any compressed model is characterized by its compression level per unit (e.g. per layer). With $n$ units, each available in $m$ compression levels, our objective is to find

$$\hat{M}^* = \arg\max_{v \in [m]^n} f(\hat{M}_v) \quad \text{subject to} \quad g(\hat{M}_v) \leq C,$$

where we are searching over the set of $n$-tuples over $[m]$. Assessing the performance of a model in practice typically involves benchmark tasks, which have limited scope and require lengthy evaluation. We address these challenges by using the base model as the gold standard and focusing solely on the relative degradation of our compressed models. To quantify this degradation, we measure the Kullback-Leibler (KL) divergence between the two models, as it has proven particularly robust with limited data. Empirically, we observed that already around 64K tokens of calibration data (corresponding to 8 full sample sequences for Llama-3-8B) are sufficient to reliably determine the quality of the lightweight model. To avoid confusion, we will refrain from inverting the fitness function and from now on consider the minimization problem

$$\hat{M}^* = \arg\min_{v \in [m]^n} D_{KL}(P_M \parallel Q_{\hat{M}_v}) \text{ subject to } g(\hat{M}_v) \leq C,$$

where we speak of *higher fitness* whenever the KL-Divergence is *lower*.

**Algorithm.** EvoPress starts from upon the classic $(1 + \lambda)$-evolutionary algorithm, which maintains a single search point at any given time. In each generation, $\lambda$ offspring are generated by copying the parent and then applying a mutation operator to each copy. The offspring are then evaluated on the fitness function, and the fittest one is selected. As an *elitist* evolutionary algorithm, the $(1 + \lambda)$-EA replaces its parent only if the best offspring has superior fitness.

We change this standard algorithm in two important ways. The first is by introducing *level-switch mutation*, a simple mutation operator that ensures high locality while preserving the compression constraint. The operator involves first

---

**Algorithm 1:** EvoPress: A $(1 + \lambda)$-Evolutionary Algorithm with Level-Switch Mutation and Multi-Step Selection for Maximizing $f : [m]^n \to \mathbb{R}$.

---
**Initialization:** candidates $\leftarrow$ [] ;
**for** $i \leftarrow 1$ **to** *initialCandidates* **do**
  candidate $\leftarrow$ sampleUniformly();
  candidates.append(candidate);
$x^{(1)} \leftarrow$ selectTopKFittest(candidates, initialTokens, $K = 1$);
**Optimization: for** $t \leftarrow 1$ **to** $\infty$ **do**
  offspring $\leftarrow$ [];
  **Mutation: for** $i \leftarrow 1$ **to** $\lambda$ **do**
    $y_i \leftarrow x^{(t)}$;
    $y_i \leftarrow$ LevelSwitchMutation($y_i$);
    offspring.append($y_i$);
  **Selection: for** $step \leftarrow 1$ **to** *selectSteps* **do**
    **Elitism: if** $step = selectSteps$ **then**
      offspring.append($x^{(t)}$);
    offs. $\leftarrow$ selectTopKFittest(offs., tokens[step], $K = $ survivors[step]);
  $x^{(t+1)} \leftarrow$ offspring[0];

---

randomly selecting one unit and increasing its compression level. Next, a second unit is sampled until one with a matching level step size is found, and its compression level is decreased. This approach ensures that 1) the compression constraint is preserved, and 2) the offspring model maintains high similarity to the parent model – an important feature for achieving rapid convergence.

The second modification is that we employ a very aggressive form of *multi-step selection*. In the first stage, all $\lambda$ offspring are evaluated using only a fraction of a full sample. From this, only a small subset of the fittest offspring are selected to compete in the next stage, where they are evaluated on a significantly larger sample size. This process is repeated once more, and in the final stage, the few remaining offspring are evaluated against the parent using a "full" minibatch, consisting of approximately 20-50 times the number of tokens used in the first stage.

For initialization, we apply the target level directly if it matches an available setting (e.g., all layers at 70% sparsity for an average of 70% sparsity). If the target falls between two compression levels (e.g., for block dropping), we initialize by randomly sampling candidates with some units compressed to the next lower level, and others to the next higher level, selecting the fittest among them. A high level overview of this procedure can be found in Algorithm 1.

**Design Considerations.** Randomized search heuristics are heavily influenced by the exploration-exploitation dilemma, i.e. the trade-off between exploring a broader solution space and intensifying the search around the currently-

best solutions. Many applications utilize sophisticated search procedures, such as genetic algorithms, to enhance exploration, that often maintain a large population, introduce crossover operations, and adopt non-elitist strategies, where parents have no chance of survival into the next generation. However, implementing these approaches for LLM compression would come with significant computational costs.

Crossover, for instance, is only effective if population diversity is preserved, for example measured by the sum of pairwise Hamming distances between individuals (Jansen & Wegener, 2002; Opris et al., 2024). While this promotes a more thorough exploration of the search space, it requires allocating resources to less promising regions, which slows down the progress toward optimal solutions. Similarly, non-elitist algorithms, despite their ability to escape local optima (Dang et al., 2021; Jorritsma et al., 2023; Lengler et al., 2024), also incur costs by frequently discarding potentially useful individuals. Consequently, these approaches should be reserved for situations where the fitness landscape is rugged, and escaping local optima is critical to finding better solutions.

**Convergence.** Contrary to many real-world problems, dynamic model compression with a carefully designed level database induces a notably smooth fitness environment, where small changes in the compressed model tend to lead to small changes in performance. A key insight into the effectiveness of evolutionary approaches is that, although the search space expands exponentially with the number of units considered, the maximum Hamming distance between any two search points in the search space increases only linearly. Therefore, as long as we receive a "signal" indicating the direction of improvement, even with seemingly limited progress per generation, we can converge rapidly to a high-quality solution.

To illustrate this, we consider the problem of removing pairs of consecutive blocks of Llama-3-8B. We perform a brute-force search over all possible 8008 block removal configurations, where six pairs of blocks are removed. Our method identifies the optimal configuration by the 6th generation, having evaluated only 16 candidates for initialization and 8 candidates per generation. Figure 1 illustrates how the algorithm approaches the optimum in Hamming distance.

Consequently, EvoPress is heavily exploitation-focused: we rely on elitism, introduce minimal mutation, maintain only a single offspring, and therefore employ zero population diversity. We present ablations and a short discussion on these choices in Appendix B. EvoPress excels at optimizing smooth fitness environments, a capability we theoretically support by proving rapid convergence under an $\ell_1$-constraint for the class of linear functions. (Here, one bit corresponds to two compression levels, while each weight of the linear function corresponds to the "saliency". The $\ell_1$-constraint is now equivalent to a compression constraint.)

**Theorem 3.1.** *Let $n, k \in \mathbb{N}$ with $k \leq n$ and consider the $(1 + \lambda)$-EA with $\lambda \in O(n/\log(n))$ and level-switch mutation. Then any linear fitness function $f : \{\mathbf{x} \mid \mathbf{x} \in \{0,1\}^n, \|\mathbf{x}\|_1 = n - k\} \to \mathbb{R}$ is optimized in expected*

$$O\left(k \cdot (n - k) \cdot \frac{1}{\lambda}\right) \text{ generations.}$$

**Discussion.** The proof is based on stochastic drift analysis and can be found in Appendix A. Notably, by increasing the number of offspring per generation, we can reduce the number of generations required for convergence, with the reduction scaling proportionally to $\lambda$ up to a reasonably large value. Since our approach uses a highly aggressive form of multi-step selection, the benefit is not simply a zero-sum trade-off. Evaluating many offspring in each generation incurs a significantly lower per-offspring computational cost, leading to a substantial speedup in convergence time. This makes the algorithm highly efficient in relatively smooth fitness environments.

## 4. Experiments

We now validate the effectiveness of EvoPress for determining the optimal layer-wise compression across three approaches: (1) **layer dropping**, where the goal is to isolate the "optimal" set of blocks to drop given a target ratio, (2) **non-uniform unstructured sparsity** and (3) **non-uniform quantization**, where we are given a set of compression options per layer (sparsities or bit-widths), and the goal is to find the "optimal" configuration that matches a certain model size. We focus on LLM compression, given the major interest in the reduction of their model size and inference latency, but our method is general and can be applied to any neural network architecture and application domain.

**Experimental Setup.** We consider base models from the Llama-2 and Llama-3 (Touvron et al., 2023) families, Mistral-v0.3 (Jiang et al., 2023), and the instruction-tuned Phi3-Medium-instruct-128k model (Abdin et al., 2024). We adopt KL-divergence as our fitness function as it provides a stronger and more robust signal compared to perplexity, reflecting the predictive distribution of the original model. We present ablations to validate this choice in Appendix B.3.

Concretely, our algorithm works as follows: Initially, for the case of quantization between available bit widths (e.g. 2.5 bit) and block dropping, we produce a number of initial configurations (around 32), evaluate them on a few data samples, and take the fittest one. For quantization with available target bitwidth and unstructured sparsity, we simply initialize using the uniform configuration. Then, we generate

new offspring in each generation by making a small number of random switches in compression levels, where the number of switches is sampled from `min(randint(1,3), randint(1,3))` and compression levels are exchanged in such a way that the overall compression ratio is maintained. We perform selection in multiple steps by iteratively choosing only the best configurations for survival, where each round uses progressively more tokens and has fewer survivors. To ensure elitism, we add the current parent to the candidate pool in the last stage of selection. Finally, after two or three of such stages, we take the last remaining configuration and adopt it as the population for the next round. We selected the number of generations, offspring, and tokens based on the search space size but found the search to be highly robust. Even drastic changes in hyperparameter settings yield similar results (see, e.g., Figure 3). The detailed parameter setting is described in Appendix C.1. For our main results we used a fixed number of generations for optimization, which was chosen conservatively to better understand convergence behavior. In practice, stopping the search early can significantly reduce runtime requirements. We discuss a simple stopping criteria and provide runtime comparisons in Appendix G.

To perform per-layer compression via unstructured sparsity and quantization, we adopt the data-aware compression methods SparseGPT (Frantar & Alistarh, 2023) and GPTQ (Frantar et al., 2022). For this purpose, we use Fineweb-Edu (Penedo et al., 2024) as a source of clean and diverse calibration data. Following Egiazarian et al. (2024), we fix the total number of calibration tokens to 8 million (8M). For a fair comparison, all competitive methods employ the same calibration data.

**Evaluation.** We follow a standard evaluation protocol (Frantar et al., 2022), measuring perplexity on the WikiText-2 (Merity et al., 2016) and C4 (Raffel et al., 2019) datasets for language performance and accuracy on zero-shot evaluations on standard benchmarks: WinoGrande (Sakaguchi et al., 2021), PiQA (Tata & Patel, 2003), HellaSwag (Zellers et al., 2019), ARC-easy and ARC-challenge (Clark et al., 2018) via the LM Eval Harness (Gao et al., 2021).

### 4.1. Application 1: Depth Pruning

We first apply EvoPress on Depth Pruning. Although removing entire transformer blocks generally results in large accuracy losses, this approach recently attracted attention in the context of initializing smaller models, as it guarantees speedups proportional to the sparsity (Samragh et al., 2023; Kim et al., 2024). Additionally, block dropping provides insights into the capabilities of transformer models, making it relevant for interpretability. We will compare against the following baselines: (1) **Shortened Llama** (Kim et al.,

2024), which scores blocks on the perplexity change after removal; (2) **ShortGPT** (Men et al., 2024), where blocks are scored based on the average cosine similarity between input and output embeddings, including the residual stream; (3) **Weight Subcloning** (Samragh et al., 2023), where blocks are scored using the ratio $||f(x)||/||f(x) + x||$, where $x$ is the input embedding and $f(x)$ is the block's output, excluding the residual stream; (4) **Sliding Window Cosine Similarity** (Gromov et al., 2024), where sets of consecutive blocks are scored based on the cosine similarity between embeddings before and after the blocks, including the residual stream. While Gromov et al. (2024) directly score entire removal configurations, the other approaches determine block removals based on their isolated scores.

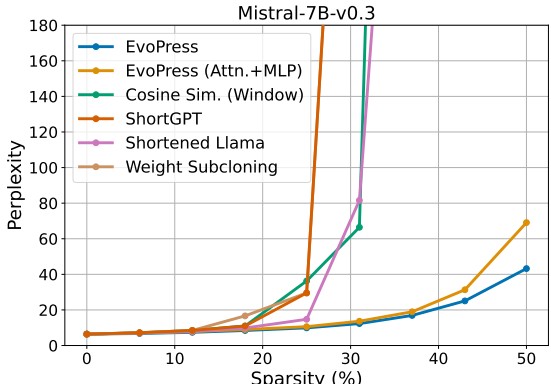

Figure 2. Depth pruning results, on Mistral-7B-v0.3. Relative to all prior methods, EvoPress shows significantly lower PPL gap relative to the uncompressed model, with remarkably large gaps at medium compression rates.

**Search Space.** In our approach, attention and MLP modules are treated independently rather than as a single unit. For each module, there are two options: either retain it or remove it. To achieve a target sparsity/depth, we initially remove an equal number of attention and MLP modules. During mutation, we allow compression level adjustments only between modules of the same type. We leave it open for future research to remove this constraint to allow flexibility in the number of removed attention and MLP modules.

**Results.** Figure 2 compares our method with baselines from previous work on Mistral-7B-v0.3. For a better comparison, we also included results where only entire transformer blocks are removed (Attn.+MLP). EvoPress consistently outperforms all previous methods, showing significant improvements even at medium sparsity levels. While all baseline methods fail entirely beyond 31.25% sparsity, EvoPress identifies functional submodels even when removing half of the model. To our knowledge, this is the first method to achieve such results. We observed similar collapses in Llama-2-7B, Llama-3-8B and Llama-3.1-8B. Overall, Evo-

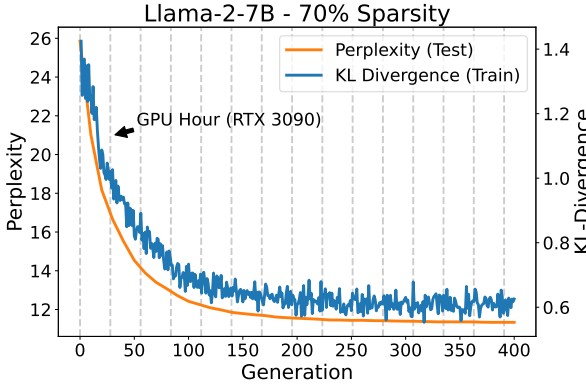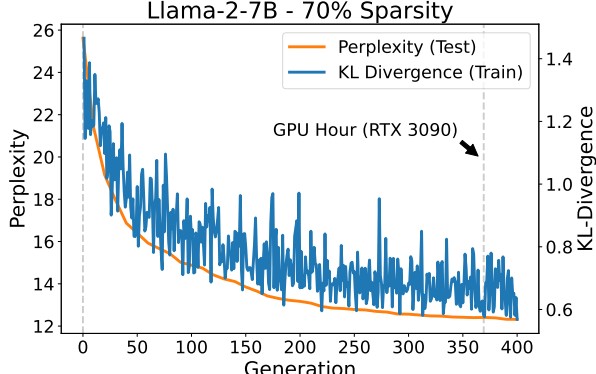

*Figure 3.* **Left**: The convergence of EvoPress vs. number of generations and wall-clock time (on a single RTX 3090 GPU with 24GB RAM) for Llama-2-7B. We observe convergence close to optimum in 5-6h; **Right**: Convergence of the "super-fast" version which reduces the number of tokens used for each evaluation. It converges to similar accuracy in little over one hour, in the same setting. The KL-Divergence corresponds to the fitness of the survivor in each generation, which is measured on a random minibatch of the entire training dataset. The perplexity is computed on the entire test dataset.

Press consistently outperforms all baselines across all tested models and sparsities (see Appendix D.1 for full results), and does so in a matter of minutes (Appendix D.2). We provide runtime comparisons as well as additional comparisons to the iterative search methods SLEB (Song et al., 2024) and BlockPruner (Zhong et al., 2024) in Appendix G.

All four previous methods rely on human-crafted scoring methods to identify the optimal combination of transformer blocks to remove. This is not only suboptimal, but also prone to bias, as their results may reflect the characteristics of the method itself rather than the model's true behavior. Specifically, we found that most scoring methods tend to favor deeper blocks, resulting in highly similar removal configurations across different prior scoring methods (Appendix Table 13). This likely occurs because methods that bias towards deeper blocks generally perform better than those that focus on earlier blocks, although neither may be optimal. In contrast, EvoPress employs an unbiased approach, offering more accurate and meaningful insights into the model.

### 4.2. Application 2: Unstructured Sparsity

Next, we examine performance for *unstructured sparsity*, which offers more fine-grained compression. The standard approach is to allocate sparsity *uniformly across layers*. However, some layers may be more sensitive to sparsity, which can significantly impact the model's output. To address this, OWL (Yin et al., 2024) introduces the Layer Outlier Distribution (LOD) metric as a measure of layer saliency, and computes a sparsity profile that is weighted by LOD. We compare EvoPress with both uniform sparsity and OWL. For OWL we used the same hyperparameter grid as the original work and took the configuration yielding the best perplexity for each model.

**Search Space.** Sparsity levels are generated as follows: For each layer, we first produce the base level corresponding to the targeted average sparsity. Then, we generate both higher and lower compression levels, where the difference between two levels corresponds to a fixed number of weights. In our experiments, we used a "step size" of 1M weights uniformly. This approach enables the mutation of compression levels across all layers, independently of their size. We adopt SparseGPT (Frantar & Alistarh, 2023) for layer pruning. We provide results of EvoPress with Wanda pruning (Sun et al., 2024) in Appendix G.

*Table 2.* Performance of various methods at 70% average sparsity. EvoPress outperforms prior methods both in terms of validation perplexity (PPL) and zero-shot accuracy.

| Model | Method | Wiki2↓ | C4↓ | Task Avg.↑ |
|---|---|---|---|---|
| Mistral-7B-v0.3 | Dense | 4.82 | 7.72 | 68.7 |
| | Uniform | 23.08 | 30.03 | 49.9 |
| | OWL | 17.22 | 21.66 | 51.9 |
| | EvoPress | **14.42** | **16.46** | **53.8** |
| Llama-3-8B | Dense | 5.54 | 7.10 | 68.6 |
| | Uniform | 85.84 | 98.35 | 44.1 |
| | OWL | 48.07 | 52.32 | 48.4 |
| | EvoPress | **28.76** | **33.72** | **50.8** |
| Phi-3-Medium-14B | Dense | 4.02 | 8.31 | 73.2 |
| | Uniform | 16.66 | 24.73 | 56.5 |
| | OWL | 15.66 | 23.38 | 55.4 |
| | EvoPress | **13.83** | **19.13** | **59.8** |

**Experimental Results.** We compare different methods for pruning to 50%, 60% and 70% unstructured sparsity. We report the 70% results in Table 2; the 50% and 60% results can be found in Appendix Tables 14 and 15, respectively.

As illustrated in Table 2, EvoPress successfully finds better profiles than uniform sparsity and noticeably outperforms competitive methods on PPL and zero-shot average accuracy by large margins on all models.

Examining sparsity profiles (Appendix Figures 12 and 13), we observe that EvoPress prunes the first blocks less aggressively, blocks in the beginning of the second half of the model more aggressively while keeping the even deeper blocks relatively dense. Notably, EvoPress assigns high importance to the v_proj matrices, reducing its sparsity to below 45%, compared to an overall average of 70%.

**Running Time.** EvoPress is also time-efficient. Figure 3 illustrates the rapid convergence of our method vs. iterations and time, with steady improvements in test perplexity. Moreover, by reducing the number of tokens used in the multi-step selection evaluation, by $4\times$ in the first step and $8\times$ in the last step, and making each generation have fewer offspring, we can significantly speed up the search. This "super-fast" version converges in a little over one GPU hour to similar test PPL (Figure 3, right), demonstrating the robustness of EvoPress, which can lead to further gains.

*Table 3.* Performance of various profiles at 3 bit quantization, for PPL and avg. zero-shot accuracy.

| Model | Method | Wiki2↓ | C4↓ | Task Avg.↑ |
|---|---|---|---|---|
| Mistral-7B-v0.3 | Dense | 4.82 | 7.72 | 68.7 |
| | Uniform | 5.54 | 8.57 | 66.3 |
| | DP | 5.79 | 8.84 | 66.0 |
| | EvoPress | **5.21** | **8.42** | **67.1** |
| Llama-3-8B | Dense | 5.54 | 7.10 | 68.6 |
| | Uniform | 12.19 | 15.76 | 60.2 |
| | DP | 29.00 | 20.03 | 61.3 |
| | EvoPress | **7.49** | **12.03** | **64.3** |
| Phi-3-Medium-14B | Dense | 4.02 | 8.31 | 73.2 |
| | Uniform | 5.18 | 9.05 | 70.0 |
| | DP | 5.72 | 9.71 | 69.1 |
| | EvoPress | **5.09** | **9.00** | **70.8** |

### 4.3. Application 3: Quantization

Finally, we apply EvoPress to the more challenging task of non-uniform neural network quantization, where the widely adopted baseline is *uniform per-layer quantization* (Frantar et al., 2022; Lin et al., 2023; Chee et al., 2023). Additionally, we consider a DP-based approach for comparison. (While OWL has also been applied to quantization, the authors found that it underperforms even relative to uniform per-layer quantization (Yin et al., 2024).) The DP search is very similar to SPDY (Frantar & Alistarh, 2022), where the goal is to minimize the Normalized Mean Squared Error (NMSE), defined as $\text{NMSE} = \|\hat{Y} - Y\|_2^2 / \|Y\|_2^2$, where $Y$ represents

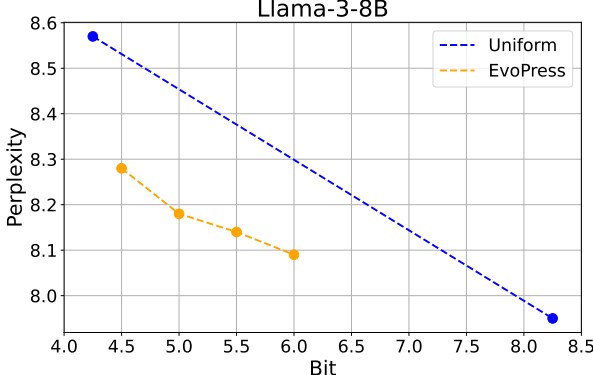

*Figure 4.* EvoPress enables non-uniform quantization with end-to-end speedups in vLLM. vLLM supports only 4-bit and 8-bit quantization, while requiring the query/key/value matrices as well as the up/gate projections to share the same bitwidth. The displayed average bitwidths include the overhead from groupwise scales.

the original model output at a layer, and $\hat{Y}$ the output of the compressed model. Then, the optimal compression allocation can be determined via a dynamic programming (DP) approach. The full SPDY method applies a second iterative random search step, which is very expensive to implement at LLM scale, and is therefore omitted.

**Search Space.** For each linear layer, we produce different configurations via GPTQ (Frantar et al., 2022) with a standard group size of 128. In each step of the evolutionary search, the bitwidth of some layers is increased while the bitwidth of others is decreased. To facilitate uniform transitions between compression levels, quantization options differ by integral bits (1 bit in the following). Since different layers may have different sizes, we allow swaps only between projections with the same number of weights.

**Experimental Results.** To validate the effectiveness of EvoPress, we consider the challenging problem of quantization to 3 bits and below. For this compression rate, uniform GPTQ quantization faces significant performance drops, motivating a dynamic quantization bitwidth allocation. We produce quantization levels at 2, 3, 4, 5, and 6 bits and search for an optimal compression profile with respect to the fitness function. The results in Table 3 indicate that non-uniform quantization with EvoPress produces superior models compared to the baseline methods. The improvements are even more pronounced at 2.25 bit and 2.5 bit quantization, as detailed in Appendix F.1. To show the real-world applicability of EvoPress, we additionally tested a restricted search limited to vLLM-supported options. vLLM allows only 4-bit and 8-bit quantization, and requires uniform bitwidth for query/key/values matrices and up/gate projections within a block. Figure 4 demonstrates strong

mixed precision quantization even under these constraints.

We visualize the configurations found by EvoPress for Llama-3.1-8B in Appendix Figures 15 and 16. We observe that the second and final blocks are compressed less aggressively, while the first block undergoes the highest compression. This contrasts with unstructured sparsity, where the first block is among the least compressed. Therefore, dynamic compression allocation must account for the specific compression method used, which underscores the importance of automated compression allocation.

Overall, we observe that EvoPress yields significant accuracy improvements (e.g., 4.1 on the zero-shot averages for Llama-3-8B), compared to the uniform profile. Moreover, the improvement over the next-best method is always significant, both in terms of perplexity and zero-shot accuracy.

## 5. Conclusion

We have presented EvoPress, an optimization framework for non-uniform compression. EvoPress is based on a new evolutionary search algorithm with low sample and iteration complexity, especially well-suited to loss landscapes in LLM compression. Specifically, we have shown that EvoPress can converge extremely fast to accurate configurations for various non-uniform LLM compression problems, and is also fast to execute in practice. We also emphasize the breadth of our study, our method was implemented and tested on three different compression approaches, relative to prior work which largely focused on a single application. Interesting directions we did not investigate are 1) combining *different compression approaches* into the same search space, and 2) finer-grained structured pruning. We plan to investigate this in future work.

## Impact Statement

This paper presents work whose goal is to advance the field of Machine Learning. There are many potential societal consequences of our work, none of which we feel must be specifically highlighted here.

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

# A. Convergence Proof of EvoPress

## A.1. A Warm-Up Argument for a Single Offspring

The overall goal of this section is to prove Theorem 3.1. As the main argument is quite complex, relying heavily on stochastic drift analysis, we begin with a warm-up, namely by presenting a simpler proof for the restricted case where $\lambda = 1$.

Unlike the practical application of Algorithm 1, this section assumes that each fitness evaluation returns the exact, or 'true,' fitness value, ignoring any noise introduced by minibatching. Additionally, our results hold for any initialization. To align with standard notation in the runtime analysis of evolutionary algorithms, we will count generations starting from zero (i.e., using 0-based indexing).

**Theorem A.1** (Single offspring). *Let $n, k \in \mathbb{N}$ with $k \leq n$ and consider the $(1 + 1)$-EA with level-switch mutation. Then any linear fitness function $f : \{\mathbf{x} \mid \mathbf{x} \in \{0, 1\}^n, \|\mathbf{x}\|_1 = n - k\} \to \mathbb{R}$ is optimized in expected*

$$O(k \cdot (n - k)) \text{ generations.}$$

*Proof.* Let $w \in \mathbb{R}^n$ be the weights associated with the linear function such that $f(x) = \sum_{i=1}^{n} x_i \cdot w_i$. To derive an upper bound, we can assume that no two weights are equal[1]. Furthermore, assume without loss of generality that these weights are sorted increasingly, meaning $w_1 < w_2 < ... < w_n$, and that $k \leq (n - k)$, as the other case follows from symmetry. Since $f$ is defined on the bit strings with exactly $k$ 0's its unique optimum is now given by $x_{\text{opt}} = 0^k 1^{n-k}$. Denote by $x^{(t)}$ the search point at step $t$ and let

$$T = \inf\{t \geq 0 \mid x^{(t)} = x_{\text{opt}}\}$$

be the number of generations required until the optimum is found.
Define $X^{(t)} = \sum_{j=1}^{k} x_j^{(t)}$ as the random variable that captures the number of 1's in the first $k$ bits of the search point at step $t$. We observe the following:

1. $X^{(t)} = 0 \Leftrightarrow x^{(t)} = x_{\text{opt}}$;

2. $X^{(t)}$ is non-increasing;

3. $X^{(t)} - X^{(t+1)} \leq 1$;

4. $X^{(0)} = \sum_{j=1}^{k} x_j^{(0)}$.

It follows that given the initial search point $x^{(0)}$ we can decompose $T$ into $s = \sum_{j=1}^{k} x_j^{(0)}$ stages $T_1, T_2, ..., T_s$, where $T_j = \inf(\{t \geq 0 \mid X^{(t)} = j - 1\}) - \inf(\{t \geq 0 \mid X^{(t)} = j\})$ captures the number of generations spent at stage $j$. By the linearity of expectation, we have

$$\mathbb{E}[T \mid X^{(0)} = s] = \sum_{j=1}^{s} \mathbb{E}[T_j].$$

It remains to bound the expected time spent at each stage. Each offspring is generated by copying the parent, selecting a 1-bit uniformly at random, selecting a 0-bit uniformly at random, and finally flipping both bits. At stage $j$ exactly $j$ of the $k$ 0-bits are among the last $n - k$ positions and exactly $j$ of the $n - k$ 1-bits are among the $k$ first positions. Hence, $j^2$ out of the total $k(n - k)$ (1-bit position, 0-bit position)-pairs advance the optimization to the next stage, yielding

$$\mathbb{P}[X^{(t+1)} = j - 1 \mid X^{(t)} = j] = \frac{j^2}{k(n - k)}.$$

Therefore, $T_j \sim \text{Geometric}\left(\frac{j^2}{k(n-k)}\right)$ and

$$\mathbb{E}[T_j] = \frac{k(n - k)}{j^2}.$$

---

[1]Formally, this can be shown using stochastic domination, which involves coupling the potentials in both cases and proving that, given the same randomness, one is always at least as large as the other.

To obtain an upper bound, we can make a worst-case assumption by setting $X(x^{(0)}) = k$. We conclude

$$\mathbb{E}[T] \leq \mathbb{E}[T|X^{(0)} = k] = \sum_{j=1}^{k} \mathbb{E}[T_j] = k(n-k) \sum_{j=1}^{k} \frac{1}{j^2} \in O(k(n-k)).$$

$\square$

**Discussion.**    Observe that, under the assumption that the probability of initializing at the optimum is sufficiently small, the proof is tight up to a constant factor of 2.

It is important to note that the above proof relies on the key assumption that whenever one of the $j^2$ "good" pairs is selected during mutation, the resulting offspring is the fittest among all candidates. This condition holds naturally when there is only a single offspring, as the offspring produced by flipping one of the $j^2$ pairs will have higher fitness than the parent. However, in the case of multiple offspring, this approach breaks down, as an offspring produced by flipping one of the $j^2$ "good" pairs might still have lower fitness than another offspring that was not generated by flipping one of these $j^2$ "good" pairs.

### A.2. The Main Argument

Drift analysis, originally developed to study all kinds Markov chains, has become the most widely used technique for analyzing the runtime of evolutionary algorithms in recent years. It works by first defining a potential function $X^{(t)}$ that measures the progress over each step $t$ of the optimization. By estimating how this potential changes at each step in expectation, i.e., computing the *drift* in $X^{(t)}$, one can then make probabilistic statements about the number of steps required until the potential reaches a certain threshold, also called the *hitting time*. To this end, a variety of drift theorems have been established, two of which will be employed in our proof. For a more thorough introduction to drift analysis, we refer to Lengler (2020).

First of all, we will utilize the the Multiplicative Drift Theorem, more specifically a tail bound introduced by Doerr and Goldberg, which is applicable when the potential decreases by a constant fraction in each step.

**Theorem A.2** (Multiplicative Drift, Tail Bound (Doerr & Goldberg, 2010)). *Let $(X^{(t)})_{t \geq 0}$ be a sequence of non-negative random variables over a finite state space $S \subset \mathbb{R}_0^+$. Assume that $X^{(0)} \leq b$ and let $T$ be the random variable that denotes the first point in time $t \in \mathbb{N}$ for which $X^{(t)} \leq a$, for some $a \leq b$. Suppose that there exists $\delta > 0$ such that for all $t < T$,*

$$\mathbb{E}[X^{(t)} - X^{(t+1)} \mid X^{(t)}] \geq \delta X^{(t)}$$

*Then,*

$$\mathbb{P}[T > \frac{t + \log(b/a)}{\delta}] \leq e^{-t}.$$

Additionally, we will employ Johannsen's Variable Drift Theorem. This theorem provides more flexibility compared to the Multiplicative Drift Theorem, as it can be applied when the drift is bounded by any *increasing* function of the potential. This often occurs naturally, as optimization typically becomes more difficult approaching the optimum.

**Theorem A.3** (Variable Drift Theorem (Johannsen, 2010)). *Let $(X^{(t)})_{t \geq 0}$ be a sequence of non-negative random variables over a finite state space $S \subset \mathbb{R}_0^+$. Let $s_{min} := \min(S \setminus \{0\})$, let $T := \inf\{t \geq 0 \mid X^{(t)} = 0\}$, and for $s \in S$ let $\Delta^{(t)}(s) := \mathbb{E}[X^{(t)} - X^{(t+1)} \mid X^{(t)} = s]$. If there is an increasing function $h : \mathbb{R}^+ \to \mathbb{R}^+$ such that for all $s \in S \setminus \{0\}$ and all $t \geq 0$,*

$$\Delta^{(t)}(s) \geq h(s),$$

*then*

$$\mathbb{E}[T] \leq \frac{s_{min}}{h(s_{min})} + \mathbb{E}\left[\int_{s_{min}}^{X^{(0)}} \frac{1}{h(\sigma)} d\sigma\right],$$

*where the expectation on the latter term is over the random choice of $X^{(0)}$.*

We will first prove an auxiliary lemma, which will play a central role in bounding the drift. For this purpose, we define an *inversion* in a bit string $x \in \{0,1\}^n$ as a pair of indices $(i,j)$ such that $i < j$ and $x_i > x_j$. The distance between these indices, $j - i$, will be referred to as the *spread* of this inversion.

**Lemma A.4.** *Let $x \in \{0,1\}^n$ be an arbitrary bit string with $k$ 0-bits and denote by $s$ the number of inversions in $x$. Then, the average spread of these inversions is at least $\sqrt{s}/16$.*

*Proof.* Consider the bit string $1^{n-k}$ containing all 1-bits of $x$. We can now generate an arbitrary bit string $x \in \{0,1\}^n$ with $k$ 0-bits and $s$ inversions by adding $k$ 0-bits in such a way that $s$ inversions are generated. Observe that adding a 0-bit after the $j$'th 1-bit results in exactly $j$ additional inversions, regardless of the other 0-bits. This means that the order in which the 0-bits are added does not affect the outcome. We proceed by a case distinction depending on how the inversions are generated.

**Case 1:** at least $s/2$ inversions are generated by adding 0-bits after the $\sqrt{s}$'th 1-bit.

For each 0-bit that is added after the $\sqrt{s}$'th 1-bit, at least half of the resulting inversions have spread at least $\sqrt{s}/2$. Consequently, this implies that there are at least $s/4$ inversions having spread at least $\sqrt{s}/2$ in total.

**Case 2:** fewer than $s/2$ inversions are generated by adding 0-bits after the $\sqrt{s}$'th 1-bit.

It follow that more than $s/2$ inversions are generated by adding 0-bits not after the $\min(n - k, \sqrt{s})$'th 1-bit. Observe that each 1-bit can participate in at most $j$ inversions with spread at most $j$. More specifically, each 1-bit can be part of at most $\sqrt{s}/4$ inversions with spread at most $\sqrt{s}/4$. Because all of the $s/2$ inversions that are added contain one of the first $\min(n - k, \sqrt{s})$ 1-bits, at most $s/4$ of these inversions can have spread at most $\sqrt{s}/4$. Therefore, we conclude that the average spread of all inversions must be at least $\sqrt{s}/16$. $\qquad\square$

We continue to prove the final result.

**Proof of Theorem 3.1**

*Proof.* As in the proof of Theorem A.1 let $w \in \mathbb{R}^n$ represent the weights associated with a linear function of the form $f(x) = \sum_{i=1}^{n} x_i \cdot w_i$. To establish an upper bound, we can again assume that no two weights are equal. Additionally, without loss of generality, assume that the weights are ordered in increasing value, i.e., $w_1 < w_2 < \cdots < w_n$, and that $k \leq n - k$, as the other case follows by symmetry. Let $x^{(t)}$ denote the search point at step $t$, and define

$$T = \inf\{t \geq 0 \mid x^{(t)} = 0^k 1^{n-k}\}$$

as the number of generations required to reach the optimal solution.
Consider the potential function

$$X^{(t)} = \sum_{i=1}^{n} (1 - x_i^{(t)}) \cdot i - \frac{k \cdot (k+1)}{2},$$

which captures the number of inversions at step $t$. Since $x_{\text{opt}} = 0^k 1^{n-k}$ is the only bit string with $k$ 0-bits without inversions, we have $X^{(t)} = 0$ if and only if $x^{(t)} = x_{\text{opt}}$. At the same time, no bit string with $k$ 0-bits has more than $k(n - k)$ inversions, hence, $X^{(t)} \leq k(n - k)$ at all times. During mutation, each of the $\lambda$ offspring is generated independently by copying the parent $x^{(t)}$, choosing uniformly at random one of the 1-bits, choosing uniformly at random one of the 0-bits and finally flipping both bits. This flipping can also be viewed as switching both bits, so that bits "move" across the search point in consecutive generations. We will use this abstraction in a later step of the proof.

As we assume the weights to be ordered increasingly, an offspring is fitter than its parent if and only if the chosen 1-bit was to the left of the chosen 0-bit, meaning, the chosen pair during mutation was an inversion. Since there are $k(n - k)$ possible pairs in total, we have for each offspring $y_1, ..., y_\lambda$

$$\mathbb{P}[f(y_j) > f(x^{(t)}) \mid X^{(t)} = s] = \frac{s}{k(n - k)}.$$

At the same time, switching two bits corresponding to an inversion decreases the number of inversions by the difference in their positions, which we call the *spread* of an inversion. This implies that any offspring fitter than its parent must have fewer inversions than its parent and therefore, $X^{(t+1)} \leq X^{(t)}$ for all $t$. Note that we cannot make the same statement about the entire group of offspring, meaning, the fittest offspring is not guaranteed to have the fewest inversions. Since $X^{(t)}$ is non-increasing we can decompose $T$ into the number of steps required until for the first time the current search point $x^{(t)}$ has at most $k(n-k)/\lambda$ inversions and the number of steps required from there until the optimum is found. By linearity of expectation

$$\mathbb{E}[T] = \mathbb{E}[T_1] + \mathbb{E}[T_2],$$

where

$$T_1 = \inf\{t \geq 0 \mid X^{(t)} \leq \frac{k(n-k)}{\lambda}\}$$

and

$$T_2 = \inf\{t \geq 0 \mid X^{(t)} = 0\} - \inf\{t \geq 0 \mid X^{(t)} \leq \frac{k(n-k)}{\lambda}\}.$$

In the remainder of this proof we will demonstrate that each of these two phases requires only an expected $O(k(n-k)/\lambda)$ generations.

We begin by bounding the expected number of steps until the search point has at most $k(n-k)/\lambda$ inversions. As computed previously, a single offspring is fitter than its parent with probability $\frac{s}{k(n-k)}$. Since any fitter offspring has fewer inversion than its parent, the potential decreases in a given step, if and only if, at least one of the offspring is fitter. By using that each offspring is generated independently and that $s \geq \frac{k(n-k)}{\lambda}$ for this phase we get that

$$\mathbb{P}[X^{(t+1)} < X^{(t)} \mid X^{(t)} = s] = 1 - (1 - \frac{s}{k(n-k)})^\lambda \geq 1 - e^{\frac{-\lambda s}{k(n-k)}} \geq 1 - e^{-1}.$$

This means, in phase 1 we have a constant probability of decreasing the potential every step. However, the resulting constant drift only provides an upper bound of $O(k(n-k))$ via the Additive Drift Theorem ([He & Yao](#), 2004). Improving this constant drift bound is challenging because we must establish a lower bound on the expected reduction in the number of inversions, given the existence of a fitter offspring. The number of inversions in an offspring is not independent of its fitness, and there is no guarantee that a fitter offspring will have fewer inversions than a less fit one. This issue is mitigated when there is only a single fitter offspring (as demonstrated in the proof of phase 2), but it becomes problematic when multiple offspring are fitter than the parent with high probability. For example, consider the bit string $1^1 0^{10} 1^{100} 0^1$ with corresponding weights $w_1 = 1, w_2 = 1002, w_3 = 1003, ..., w_{112} = 1112$. If $\lambda$ is reasonably large it becomes very likely that at least one of the children will have the first 1-bit chosen in mutation. This offspring is guaranteed to be the fittest one, but at the same time (assuming the chosen 0-bit is not the last one) it decreases the number of inversions very little compared to sampling one of the inversions for mutation uniformly at random. We will resolve this difficulty by a separate drift argument.

Let $B_C$ be the event that, within the next

$$\frac{2C}{1 - e^{-1}} \frac{k(n-k)}{\lambda}$$

steps, the number of inversions in $x^{(t)}$ falls below the threshold of

$$\frac{k(n-k)}{\lambda}.$$

Here, $C$ is chosen such that $\lambda \leq \frac{C}{8} \frac{n}{\log(n)}$. If we can demonstrate that $B_C$ occurs with a probability of at least some constant, then the proof of the first phase is established, as $B_C$ is expected to occur after a constant number of repetitions.

Henceforth, we will implicitly condition on $s \geq k(n-k)/\lambda$, since otherwise, the conclusion follows immediately. By the Chernoff bound over round events, the probability that the potential decreases at most

$$C\frac{k(n-k)}{\lambda}$$

times within the next

$$\frac{2C}{1 - e^{-1}} \frac{k(n-k)}{\lambda}$$

rounds is sub-constant. We will condition on the event that the potential decreases at least

$$C \frac{k(n-k)}{\lambda}$$

times, and from now on, we will only consider such potential-reducing generations.

If we regard mutation as swapping the 1-bit with the 0-bit, we can enumerate all 0-bits from 1 to $k$ and denote by $i_j$ the current position of the $j$'th 0-bit, which will be referred to as $0_j$. Note that this enumeration stays fixed across generations, meaning that the relative order can change and $0_j$ is not necessarily the $j$'th 0-bit in $x^{(t)}$. Now define

$$Z_j^{(t)} = 1 + \sum_{l=1}^{i_j} x^t$$

as the random variable that captures the number of 1-bits before $0_j$ plus one, or in other words, one plus the number of inversions this specific 0-bit is part of. Let $S_j$ denote the event that the fittest offspring was generated by a mutation that selected $0_j$ and this offspring is fitter than the parent. We continue to show that

$$\mathbb{E}[Z_j^{(t+1)} \mid Z_j^{(t)} = s, S_j] \geq \frac{s}{2}.$$

We achieve this by systematically revealing the randomness in each generation. First, uncover which 0-bit flip produced the fittest offspring[2]. Assume this bit is $0_j$. Next, reveal all offspring that were generated by flipping other 0-bits than $0_j$. Let $m$ be the number of offspring that were not uncovered yet, i.e., the number of offspring where $0_j$ was switched. Now enumerate all 1-bits to the left of $0_j$ in $x^{(t)}$ from right to left (here, relative order matters). Let $l$ be the smallest integer such that when switching the $l$'th 1-bit to the left of $0_j$ with $0_j$ the resulting offspring of $x^{(t)}$ has higher fitness than all $\lambda - m$ previously uncovered offspring. Denote by $D_l$ the corresponding event. Such $l$ must exists, since we condition on the event that some offspring with bit $0_j$ flipped (switched) is the fittest among all offspring. Because the weights are sorted increasingly it must hold that switching the $l+1$'th 1-bit with $0_j$ will also result in an offspring with higher fitness than the other $\lambda - m$ offspring, while switching the $l-1$'th 1-bit with $0_j$ will result in an offspring with lower fitness than the other $\lambda - m$ offspring. Next, uncover all offspring where bit $0_j$ was switched with one the first up to $(l-1)$'th bit to the left of $0_j$. Let $m'$ denote the number of yet uncovered offspring. Now each of the remaining $m'$ offspring is generated by flipping $0_j$ with one of the $l$'th to $(s-1)$'th 1-bits to the left of $0_j$. Observe that the fittest among them will be the one with the leftest 1-bit chosen. Therefore,

$$\mathbb{E}[Z_j^{(t+1)} \mid Z_j^{(t)} = s, S_j, D_l, m' \text{ offspring not uncovered}] = s - \mathbb{E}\left[\max_{i=1,...,m'} U_i\right],$$

where $U_i \sim \text{Uniform}(l, s-1)$. Given that we are conditioning on $S_j$, we know that the fittest offspring was produced by flipping $0_j$, which implies $m' \geq 1$. As $l \geq 1$ it follows that

$$\mathbb{E}[Z_j^{(t+1)} \mid Z_j^{(t)} = s, \ S_j] \geq s/2.$$

Denote by $\hat{T}_j$ the number of steps required until $Z_j$ reaches 1, only counting steps where $Z_j$ is decreased. Using a tail bound for the Multiplicative Drift Theorem (Theorem A.2) we have that

$$\mathbb{P}[\hat{T}_j > 2(\log(n) + \log(n-k))] \leq \frac{1}{n}.$$

As $k < (n-k)$ we conclude by a union bound that with probability at least $1/2$ each potential $Z_j$ will reach 1 within at most $4\log(n)$ steps. Therefore, with probability at least $1/2$, after $4k\log(n)$ generations where some offspring is fitter than

---

[2]More precisely, we must uncover which 0-bit flip resulted in the offspring selected during the selection process. This accounts for scenarios where multiple offspring have the same highest fitness, in which case one of the fittest candidates is typically chosen uniformly at random. As the occurrence of multiple equally fit offspring is a mere technicality, we have largely omitted further discussion of this case.

the parent, there must be 0 inversions in $x^{(t)}$. Note that in practice, there will not actually be 0 inversions in $x_t$, as the condition $s \geq k(n-k)/\lambda$ is violated earlier, leading the optimization process to enter the second phase. Using the fact that $\lambda \leq \frac{C}{8} \frac{n}{\log(n)}$ and $n - k \geq n/2$ we obtain

$$4k \log(n) \leq \frac{8k(n-k)\log(n)}{n} \leq C \frac{k(n-k)}{\lambda}.$$

Finally, as the probability of having less than $Ck(n-k)/\lambda$ "successful" generations in the considered time period is sub-constant, we conclude via another union bound that there exists a constant $C'$ such that event $B_C$ occurs with probability at least $1/C'$. Consequently, we have

$$\mathbb{E}[T_1] \leq C \cdot C' \cdot \frac{k(n-k)}{\lambda} \in O\left(\frac{k(n-k)}{\lambda}\right).$$

To compute $\mathbb{E}[T_2]$ we first bound the probability that exactly one of the generated offspring is fitter than the parent. Denote by

$$A_i = \left\{ \left| \left\{ j \in \{1, \ldots, \lambda\} \mid f(y_j) > f(x^{(t)}) \right\} \right| = i \right\}$$

the event that exactly $i$ of the offspring are fitter than the parent $x^{(t)}$. As shown earlier, the probability that a given offspring is fitter than its parent is exactly $\frac{s}{k(n-k)}$, where $s$ represents the number of inversions in $x^{(t)}$. Given that each offspring is generated independently, we have for $s \leq k(n-k)/\lambda$

$$\mathbb{P}[A_1 \mid X^{(t)} = s] = \lambda \cdot \frac{s}{k(n-k)} \cdot \left(1 - \frac{s}{k(n-k)}\right)^{\lambda - 1}$$

$$\geq \lambda \cdot \frac{s}{k(n-k)} \cdot \left(1 - \frac{s}{k(n-k)}\right)^{\frac{k(n-k)}{s} - 1}$$

$$\geq \lambda \cdot \frac{s}{k(n-k)} \cdot \frac{1}{e}.$$

Lemma A.4 indicates that when selecting an offspring uniformly at random from all those with higher fitness than the parent (i.e., those generated by flipping an inversion), the expected number of inversions in that offspring is at least $\sqrt{s}/16$ fewer than in the parent. We can now reveal the randomness in two steps. First, we only uncover how many of the generated offspring are fitter than the parent. Given that there is only a single fitter offspring, i.e., conditioned on $A_1$, we then uncover its number of inversions. Clearly, this single fitter offspring is now sampled uniformly at random from all offspring with higher fitness than $x^{(t)}$; thus, for $s \leq k(n-k)/\lambda$

$$\Delta^{(t)}(s) = \mathbb{E}[X^{(t+1)} - X^{(t)} \mid X^{(t)} = s]$$

$$= \sum_{k=0}^{\lambda} \mathbb{E}[X^{(t+1)} - X^{(t)} \mid X^{(t)} = s, A_k] \cdot \mathbb{P}[A_k \mid X^{(t)} = s]$$

$$\geq \mathbb{E}[X^{(t+1)} - X^{(t)} \mid X^{(t)} = s, A_1] \cdot \mathbb{P}[A_1 \mid X^{(t)} = s]$$

$$\geq \frac{\sqrt{s}}{16} \cdot \lambda \cdot \frac{s}{k(n-k)} \cdot \frac{1}{e}.$$

Finally, applying Johannsen's Variable Drift Theorem (Johannsen, 2010) (Theorem A.3) yields

$$\mathbb{E}[T_2] \leq 16e \frac{k(n-k)}{\lambda} + \mathbb{E}\left[\int_1^{X^{(0)}} 16e \frac{k(n-k)}{\lambda \sigma^{3/2}} \, d\sigma\right]$$

$$\leq 16e \frac{k(n-k)}{\lambda} \left(1 + \int_1^{\frac{k(n-k)}{\lambda}} \frac{1}{\sigma^{3/2}} \, d\sigma\right)$$

$$\in O\left(\frac{k(n-k)}{\lambda}\right).$$

$\square$

# B. Evolutionary Search Parameter Ablations

## B.1. Mutation Rate (Depth Pruning)

The mutation rate plays a crucial role in balancing exploration and exploitation. A higher mutation rate allows for broader exploration of the search space; however, this space grows exponentially with the number of mutations. As a result, when trying to approach the optimum in terms of Hamming distance, the proportion of "good" offspring decreases significantly with an increasing mutation rate. Consequently, in a smooth fitness landscape, we expect significantly faster optimization with a lower mutation rate.

To provide some mathematical intuition, consider optimizing over the 200-dimensional hypercube $\{0, 1\}^{200}$, where the current search point is $x^{(t)} = 0^{200}$ and the global optimum is $x_{\mathrm{opt}} = 1^{20}0^{180}$. For this illustration we use a mutation operator that randomly selects a subset of $k$ bits to flip. Flipping $k$ bits corresponds to selecting a bitstring from the $k$'th Hamming layer of $x^{(t)} = 0^{200}$ uniformly at random, where the $k$'th Hamming layer consists of all bitstrings with a Hamming distance of $k$ from $x^{(t)}$. Similarly, the Hamming ball of radius $k$ includes all bitstrings with a Hamming distance at most $k$. Assume that any bitstring closer to the optimum in terms of Hamming distance has higher fitness than our current search point[3]. The probability of improving the fitness via a mutation of $k$ bits equals the fraction of points in the $k$'th Hamming layer of $x^{(t)}$ that are also in the Hamming ball of radius 19 around $x_{\mathrm{opt}}$. When the current search point is reasonably close to the optimum, this ratio is maximized for $k = 1$. For the described setting, we can compute the probabilities of each event $A_k$, where $A_k$ represents the event that mutating $k$ bits of $x^{(t)}$ results in a decrease in the Hamming distance from $x_{\mathrm{opt}}$. These probabilities are given by:

$$\mathbb{P}[A_1] = \frac{\binom{20}{1}}{\binom{200}{1}} = 0.1 \qquad\qquad \mathbb{P}[A_2] = \frac{\binom{20}{2}}{\binom{200}{2}} \approx 0.0095$$

$$\mathbb{P}[A_3] = \frac{\binom{20}{3} + \binom{20}{2}\cdot\binom{180}{1}}{\binom{200}{3}} \approx 0.0269 \qquad\qquad \mathbb{P}[A_4] = \frac{\binom{20}{4} + \binom{20}{3}\cdot\binom{180}{1}}{\binom{200}{4}} \approx 0.0032$$

$$\mathbb{P}[A_5] = \frac{\binom{20}{5} + \binom{20}{4}\cdot\binom{180}{1} + \binom{20}{3}\cdot\binom{180}{2}}{\binom{200}{5}} \approx 0.0076 \quad \mathbb{P}[A_6] = \frac{\binom{20}{6} + \binom{20}{5}\cdot\binom{180}{1} + \binom{20}{4}\cdot\binom{180}{2}}{\binom{200}{6}} \approx 0.0010$$

Note that accounting for the potentially greater Hamming distance gained for higher mutation rates (i.e., calculating the drift in the Hamming distance) has only marginal effect. This is because for an odd number of mutations, most of the conditional probability mass is concentrated on the case where the Hamming distance is reduced by just one bit. Similarly, for an even number of mutations, most of the conditional probability mass is concentrated on cases where the reduction in the Hamming distance is only two bits. The advantage of a low mutation rate becomes even more pronounced as the search process nears the optimum. For instance, when the Hamming distance between $x^{(t)}$ and $x_{\mathrm{opt}}$ is 5, mutating a single bit results in a 16-fold greater drift in Hamming distance compared to any other mutation rate.

To study the empirical impact of the mutation rate on our search process, we tested various distributions from which the number of mutations is sampled. Table 4 illustrates the effects of these distributions for selecting the optimal twelve blocks to drop for Mistral-7B-v0.3. The results confirm our intuition: higher mutation rates generally reduce performance. However, sampling from the minimum of two uniform distributions ensures a reasonably high probability of choosing a low number of mutations. These offspring, with fewer mutations, then drive the optimization process, yielding comparably lower performance drops. Conversely, when we eliminate this sampling and instead use a high, constant mutation rate, we lose the locality that is crucial for evolutionary algorithms, leading to a significant drop in performance.

A low mutation rate carries the risk of getting trapped in local optima. However, as discussed in Section 3, we expect dynamic model compression to exhibit a smooth fitness landscape with few local optima. Moreover, fitness evaluations in our context are relatively expensive. Increasing the mutation rate would only be beneficial if the smaller search space had already been thoroughly explored. In our case, though, even a small neighborhood cannot be fully explored within a feasible time frame.

A widely used strategy for balancing the advantages and disadvantages of different mutation rates involves self-adjusting mutation rates, which have been shown to be effective both theoretically and in practice (Kern et al., 2004; Doerr et al., 2019). These methods decrease the mutation rate when progress is relatively "easy", and increase it when progress becomes difficult, offering a greater chance of escaping local optima.

---

[3]While this assumption does not hold in practice, it serves as a useful intuition in a reasonably smooth fitness landscape.

*Table 4.* Effect of varying the distribution determining the number of mutations.

| Number of Mutations | | Wiki2↓ | C4↓ | FW↓ |
|---|---|---|---|---|
| $\min(U_1, U_2),$ | $U_1, U_2 \sim U(1,3)$ | **17.52** | 21.60 | 16.79 |
| $\min(U_1, U_2),$ | $U_1, U_2 \sim U(1,7)$ | 21.49 | 22.41 | 17.65 |
| $\min(U_1, U_2),$ | $U_1, U_2 \sim U(1,15)$ | 18.65 | 22.67 | 17.63 |
| 1 | | 18.12 | **21.12** | **16.33** |
| 3 | | 22.09 | 25.42 | 19.25 |
| 7 | | 25.06 | 26.52 | 19.65 |
| 15 | | 27.01 | 28.19 | 22.03 |

## B.2. Multi-Step Selection (Unstructured Sparsity)

We will use this subsection to ablate the impact of hyperparameters for the multi-step selection, namely, the number of tokens and survivors. As discussed earlier in Section 4.2, the default hyperparameters we chose for our unstructured sparsity search were quite conservative. The following experiments will be conducted based on the "super-fast" version, which uses two steps of selection. It first generates 16 offspring, evaluates them on 512 tokens, and compares only the fittest one with the parent on another 8192 tokens.

Table 5 shows the impact of adapting the number of tokens in the first selection step. Note that reducing tokens is only reasonable up to a certain degree, as fitness evaluation has constant overhead independent of the number of tokens (e.g., for loading the levels). Table 6 ablates the number of offspring in each generation. All perplexities were measured after 400 generations.

*Table 5.* Effect of varying the number of tokens in first preselection step.

| Offspring | Stage 1: Tokens | Stage 2: Tokens | Wiki2↓ | C4↓ | FW↓ |
|---|---|---|---|---|---|
| 16 | 1024 | 8192 | 16.22 | **17.93** | **12.26** |
| 16 | 512 | 8192 | **15.87** | 18.28 | 12.38 |
| 16 | 256 | 8192 | 17.25 | 18.51 | 12.52 |
| 16 | 128 | 8192 | 16.01 | 18.99 | 12.72 |
| 16 | 64 | 8192 | 15.89 | 19.35 | 12.98 |

*Table 6.* Effect of varying the number of offspring.

| Offspring | Stage 1: Tokens | Stage 2: Tokens | Wiki2↓ | C4↓ | FW↓ |
|---|---|---|---|---|---|
| 64 | 512 | 8192 | 16.35 | 18.27 | **12.36** |
| 32 | 512 | 8192 | 16.65 | **18.22** | 12.44 |
| 16 | 512 | 8192 | **15.87** | 18.27 | 12.38 |
| 8 | 512 | 8192 | 16.37 | 18.74 | 12.64 |
| 4 | 512 | 8192 | 17.87 | 18.97 | 12.72 |

In a similar vein to the discussion in Appendix B.1, the number of offspring can also be dynamically adapted. Ideally, the number of offspring should increase to the point where the computational effort is compensated by the number of generations, as outlined in Theorem 3.1. Methods such as the Self-Adjusting $(1, \lambda)$-EA have recently gained significant theoretical interest and have been shown to automatically determine "ideal" offspring sizes on specific problems (Hevia Fajardo & Sudholt, 2021; Kaufmann et al., 2022).

## B.3. Fitness Environment (Quantization)

We explored an alternative fitness function by testing perplexity as opposed to KL-Divergence. One advantage of using perplexity is the reduced memory requirement, as it does not necessitate storing the logits, which can be particularly burdensome for large vocabularies. However, perplexity relies solely on the information from the ground truth token, while KL-Divergence takes into account the entire distribution. This distinction is significant only if the selection decisions vary between the two metrics. Generally, we expect KL-Divergence to perform at least as well as perplexity; however, in many

instances, their performances are similar. This observation could indicate that KL-Divergence might be using more tokens than necessary to assess fitness effectively. Although in the context of quantization KL-Divergence yielded slightly better results (Table 7, Figure 5 left), both metrics showed comparable performance when applied to unstructured sparsity (Figure 5 right).

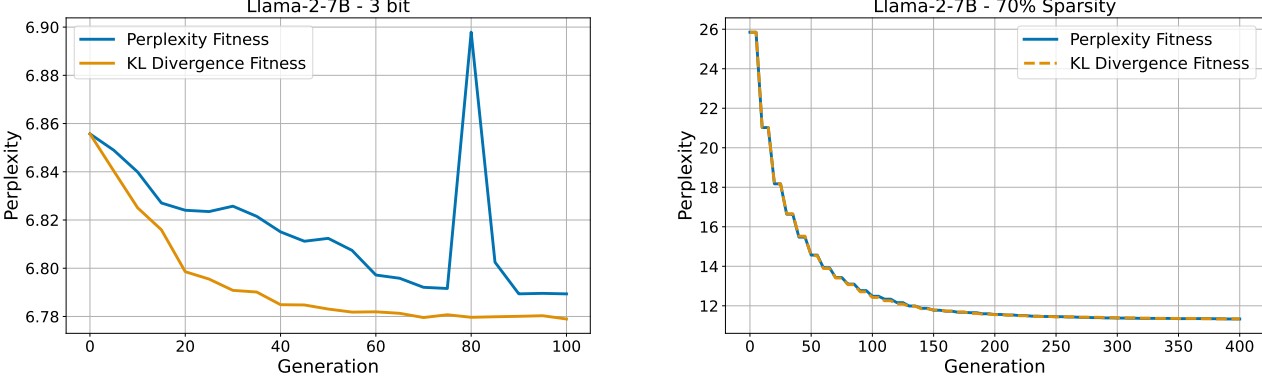

*Figure 5.* Convergence of EvoPress for unstructured sparsity (left) and quantization (right) for different fitness functions.

*Table 7.* Comparison of using KL-Divergence vs. Perplexity as fitness function.

| Model | # Bits | Method | Wiki2↓ | C4↓ | FW↓ |
|-------|--------|--------|--------|-----|-----|
| Llama-3-8B | 3 | Uniform | 12.19 | 15.76 | 11.47 |
| | | EvoPress (PPL) | 8.17 | 12.15 | 9.64 |
| | | EvoPress (KL) | **7.49** | **12.03** | **9.56** |
| | 4 | Uniform | 6.48 | 9.50 | 8.46 |
| | | EvoPress (PPL) | **5.86** | 9.46 | 8.23 |
| | | EvoPress (KL) | **5.86** | **9.44** | **8.22** |
| Llama-2-7B | 3 | Uniform | 6.16 | 7.96 | 6.86 |
| | | EvoPress (PPL) | 5.74 | 7.90 | 6.79 |
| | | EvoPress (KL) | **5.70** | **7.87** | **6.76** |
| | 4 | Uniform | 5.48 | 7.10 | 6.40 |
| | | EvoPress (PPL) | 5.25 | 7.09 | 6.37 |
| | | EvoPress (KL) | **5.22** | **7.07** | **6.34** |
| Mistral-7B-v0.3 | 3 | Uniform | 5.54 | 8.57 | 6.96 |
| | | EvoPress (PPL) | 5.23 | 8.45 | 6.87 |
| | | EvoPress (KL) | **5.21** | **8.42** | **6.86** |
| | 4 | Uniform | 5.10 | 7.87 | 6.50 |
| | | EvoPress (PPL) | 4.85 | 7.86 | 6.49 |
| | | EvoPress (KL) | **4.84** | **7.84** | **6.48** |

## C. Experimental Setup

### C.1. Hyperparameter Setting

Here, we provide an overview of the hyperparameters used in our experiments. As shown in Table 8, we employed different choices for the number of tokens, offspring, and generations for different applications to account for the size of the respective search space. However, the search is very robust with respect to these choices, and using one set of hyperparameters for another application yields similar results. This is also demonstrated through the "super-fast" version, which uses drastically different hyperparameters, but achieves comparable performance (see Figure 3 in main text).

Across all applications, we sampled the number of mutations from the distributions $\min(U_1, U_2)$ with $U_1, U_2 \sim Unif(1, 3)$, which closely mimics the behavior of using only one mutation (see the ablation study in Appendix 4).

For *Depth Pruning*, where each block has only two choices and significantly fewer blocks are present compared to layers in other methods, we leveraged the insight from Theorem 3.1, which suggests that the number of required generations scales proportionally to $k(n - k)$, where $k$ represents the number of removed blocks and $n$ the total number of blocks.

For *Unstructured Sparsity*, the search space is considerably larger, with more than 10 choices per layer[4]. As a result, more generations are necessary to converge because each generation only makes small improvement in terms of Hamming distance from the optimum.

For *Quantization*, the search space is somewhat smaller since fewer "natural" compression levels are available. However, the fitness landscape is less smooth, with significantly larger step sizes in compression levels, motivating the use of a higher number of tokens.

For all these applications, we adopted a conservative approach to the number of generations to better understand convergence. In practice, we need significantly fewer generations to converge close to optimum, as demonstrated in Section 4.2, Appendix D.2, Appendix F.2, and Appendix B.3. Additionally, we showed a much faster version (in terms of time per iteration) that uses significantly less tokens and has fewer offspring.

*Table 8.* Employed hyperparameters for different applications.

| Application | Generations | Offspring | Survivors (1) | Tokens (1) | Survivors (2) | Tokens (2) | Survivors (3) | Tokens (3) |
|---|---|---|---|---|---|---|---|---|
| Depth Pruning | $k(n-k)/1.5$ | 32 | 2 | 2048 | 1 | 32768 | N/A | N/A |
| Unstr. Sparsity | 400 | 64 | 8 | 2048 | 2 | 16384 | 1 | 65536 |
| Quantization | 150 | 128 | 16 | 2048 | 4 | 16384 | 1 | 131072 |
| Super-Fast | 400 | 16 | 1 | 512 | 1 | 8192 | N/A | N/A |

### C.2. Robustness to Random Seed

To evaluate the robustness of EvoPress, we conducted 16 independent runs with different random seeds. Specifically, we used the "super-fast" variant to determine the optimal compression allocation for Llama-3-8B at 70% sparsity, assessing perplexity scores on the C4, Wikitext2, and hold-out Fineweb-Edu datasets. The results indicate that EvoPress is highly robust, as reflected by the low standard deviation observed across the hold-out metrics (Figure 7). For example, after 1000 generations of the "super-fast" variant, the configurations found achieve a mean C4 perplexity of 33.82 with a standard deviation of 0.61, compared to 52.32 for the next best method, OWL. This improvements achieved by EvoPress are therefore statistically relevant. Furthermore, as shown in Figure 6, the configurations identified across different runs are highly similar, which is expected to improve further with additional generations.

---

[4]If needed, one could increase the step size and reduce the number of compression levels to load.

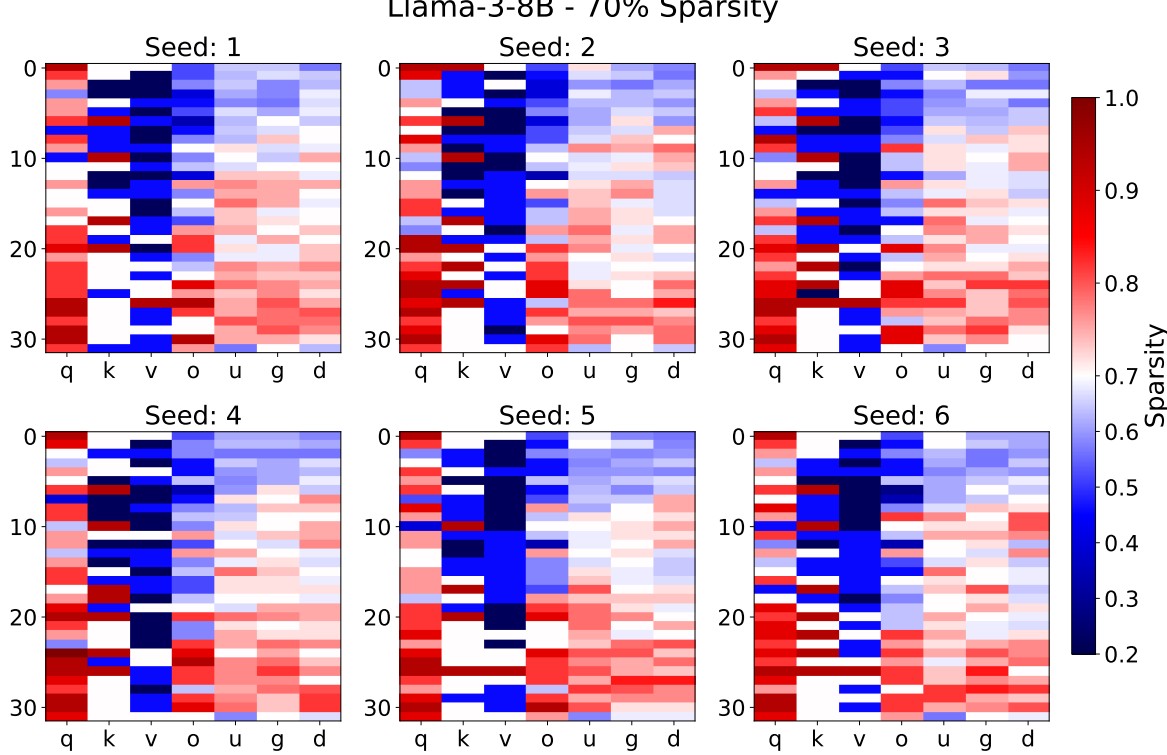

*Figure 6.* Configurations identified by EvoPress on Llama-3-8B after 1000 generations show high similarity across different seeds. The y-axis represents the depth of the respective transformer block, while the x-axis denotes the corresponding layer (q: query, k: key, v: value, o: output, u: MLP up, g: MLP gate, d: MLP down).

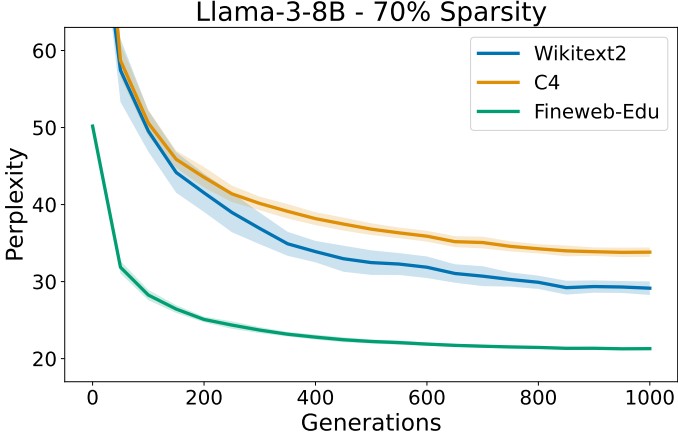

*Figure 7.* Convergence behavior of the "super-fast" variant across 16 independent runs. The extremely low standard deviation (shaded area) demonstrates the robustness of the method, which suggests that local optima do not pose significant challenges to the search.

# D. Additional Depth Pruning Results

## D.1. Full Results

Here, we present our additional results for depth pruning experiments on Mistral-7B-v0.3 (Table 12), Llama-2-7B (Table 9), Llama-3-8B (Table 10), and Llama-3.1-8B (Table 11). Across all levels of sparsities, EvoPress consistently outperforms previous methods. Additionally, Table 12 includes results where only entire transformer blocks are removed by EvoPress. This showcases that the significant gains are not primarily due to this relaxation, and that our method performs better than baselines even when dealing with this coarser search space.

*Table 9.* Depth pruning of Llama-2-7B.

| Sparsity | Method | Wiki2↓ | C4↓ | FW↓ |
|---|---|---|---|---|
| 0% | Dense | 5.21 | 6.93 | 6.40 |
| 12.5% | EvoPress | **6.42** | **8.60** | **7.54** |
| | ShortGPT | 8.86 | 10.78 | 9.30 |
| | Cosine Similarity (Window) | 7.53 | 9.82 | 8.51 |
| | Weight Subcloning | 9.09 | 11.06 | 9.60 |
| | ShortenedLlama | 7.68 | 10.44 | 8.57 |
| 25% | EvoPress | **9.15** | **11.46** | **9.69** |
| | ShortGPT | 23.41 | 30.30 | 21.16 |
| | Cosine Similarity (Window) | 16.60 | 21.04 | 17.37 |
| | Weight Subcloning | 23.41 | 30.30 | 21.16 |
| | Shortened Llama | 13.86 | 14.08 | 11.81 |
| 37.5% | EvoPress | **17.98** | **18.91** | **15.53** |
| | ShortGPT | 70.94 | 63.51 | 54.07 |
| | Cosine Similarity (Window) | 192.07 | 212.60 | 151.10 |
| | Weight Subcloning | 70.94 | 63.51 | 54.07 |
| | Shortened Llama | 35.37 | 26.07 | 20.37 |
| 50% | EvoPress | **48.84** | **42.29** | **33.57** |
| | ShortGPT | 226.14 | 171.04 | 180.51 |
| | Cosine Similarity (Window) | 4570.15 | 2876.83 | 1861.06 |
| | Weight Subcloning | 226.14 | 171.04 | 180.51 |
| | Shortened Llama | 145.78 | 87.40 | 68.79 |

*Table 10.* Depth pruning of Llama-3-8B.

| Sparsity | Method | Wiki2↓ | C4↓ | FW↓ |
|---|---|---|---|---|
| 0% | Dense | 5.54 | 8.80 | 7.62 |
| 12.5% | EvoPress | **7.72** | **12.61** | **10.15** |
| | ShortGPT | 13.21 | 19.56 | 14.25 |
| | Cosine Similarity (Window) | 9.54 | 14.87 | 11.64 |
| | Weight Subcloning | 13.21 | 19.56 | 14.25 |
| | Shortened Llama | 9.42 | 15.09 | 11.57 |
| 25% | EvoPress | **13.99** | 22.83 | **15.84** |
| | ShortGPT | 5527.54 | 11589.93 | 2346.13 |
| | Cosine Similarity (Window) | 5519.95 | 11629.61 | 2342.91 |
| | Weight Subcloning | 5527.54 | 11589.93 | 2346.13 |
| | Shortened Llama | 16.59 | **20.81** | 16.28 |
| 37.5% | EvoPress | **27.56** | **35.70** | **26.77** |
| | ShortGPT | 64281.36 | 13836.12 | 3789.09 |
| | Cosine Similarity (Window) | 64627.29 | 13890.14 | 3784.72 |
| | Weight Subcloning | 64381.36 | 13836.13 | 3789.09 |
| | Shortened Llama | 50.20 | 61.56 | 37.40 |
| 50% | EvoPress | **84.99** | **87.86** | **66.41** |
| | ShortGPT | 1663.97 | 1740.04 | 1588.20 |
| | Cosine Similarity (Window) | 2053.19 | 1116.47 | 694.00 |
| | Weight Subcloning | 1663.97 | 1740.04 | 1588.20 |
| | Shortened Llama | 724.86 | 666.41 | 210.30 |

*Table 11.* Depth pruning of Llama-3.1-8B.

| Sparsity | Method | Wiki2↓ | C4↓ | FW↓ |
|---|---|---|---|---|
| 0% | Dense | 5.61 | 8.90 | 7.67 |
| 12.5% | EvoPress | **7.58** | **12.24** | **10.00** |
| | ShortGPT | 12.54 | 19.21 | 13.76 |
| | Cosine Similarity (Window) | 12.54 | 19.21 | 13.76 |
| | Weight Subcloning | 12.54 | 19.21 | 13.76 |
| | Shortened Llama | 9.27 | 14.80 | 11.21 |
| 25% | EvoPress | **11.59** | **17.84** | **13.96** |
| | ShortGPT | 4278.39 | 6754.92 | 1512.39 |
| | Cosine Similarity (Window) | 4278.39 | 6754.92 | 1512.39 |
| | Weight Subcloning | 4278.39 | 6754.92 | 1512.39 |
| | Shortened Llama | 20.41 | 20.33 | 16.12 |
| 37.5% | EvoPress | **24.98** | **35.77** | **25.93** |
| | ShortGPT | 123044.19 | 22071.51 | 6059.03 |
| | Cosine Similarity (Window) | 123044.19 | 22071.51 | 6059.03 |
| | Weight Subcloning | 123044.19 | 22071.51 | 6059.03 |
| | Shortened Llama | 41.34 | 43.53 | 31.00 |
| 50% | EvoPress | **105.84** | **110.69** | **61.25** |
| | ShortGPT | 1630.11 | 1680.21 | 1698.64 |
| | Cosine Similarity (Window) | 1881.54 | 1196.63 | 683.24 |
| | Weight Subcloning | 1630.11 | 1680.21 | 1698.64 |
| | Shortened Llama | 454.96 | 309.42 | 153.96 |

*Table 12.* Depth pruning of Mistral-7B-v0.3.

| Sparsity | Method | Wiki2↓ | C4↓ | FW↓ |
|---|---|---|---|---|
| 0% | Dense | 4.82 | 7.72 | 6.41 |
| 12.5% | EvoPress | **6.06** | **9.00** | **7.42** |
| | EvoPress (Attn.+MLP) | 6.33 | 9.44 | 7.80 |
| | ShortGPT | 7.19 | 10.18 | 8.46 |
| | Cosine Similarity (Window) | 7.19 | 10.18 | 8.46 |
| | Weight Subcloning | 7.19 | 10.18 | 8.46 |
| | Shortened Llama | 6.64 | 9.71 | 7.94 |
| 25% | EvoPress | **8.66** | **12.04** | **9.92** |
| | EvoPress (Attn.+MLP) | 9.46 | 13.02 | 10.59 |
| | ShortGPT | 43.26 | 40.16 | 29.54 |
| | Cosine Similarity (Window) | 33.75 | 54.07 | 36.26 |
| | Weight Subcloning | 43.26 | 40.16 | 29.54 |
| | Shortened Llama | 14.94 | 19.30 | 14.73 |
| 37.5% | EvoPress | **17.52** | **21.60** | **16.90** |
| | EvoPress (Attn.+MLP) | 21.62 | 25.17 | 18.97 |
| | ShortGPT | 2898.98 | 2722.66 | 981.99 |
| | Cosine Similarity (Window) | 1034.09 | 2471.86 | 1050.56 |
| | Weight Subcloning | 2898.98 | 2722.66 | 981.99 |
| | Shortened Llama | 440.20 | 442.09 | 486.15 |
| 50% | EvoPress | **61.75** | **54.15** | **43.23** |
| | EvoPress (Attn.+MLP) | 108.91 | 99.74 | 69.07 |
| | ShortGPT | 2422.72 | 2134.92 | 1083.51 |
| | Cosine Similarity (Window) | 3411.47 | 1934.16 | 1740.91 |
| | Weight Subcloning | 2422.72 | 2134.92 | 1083.51 |
| | Shortened Llama | 5241.76 | 3595.71 | 1953.14 |

## D.2. Practical Convergence

EvoPress identifies superior compression profiles in a highly efficient manner. Figure 8 displays that the evolutionary search produces better compressed models than previous techniques in a matter of minutes, with full convergence in around half an hour.

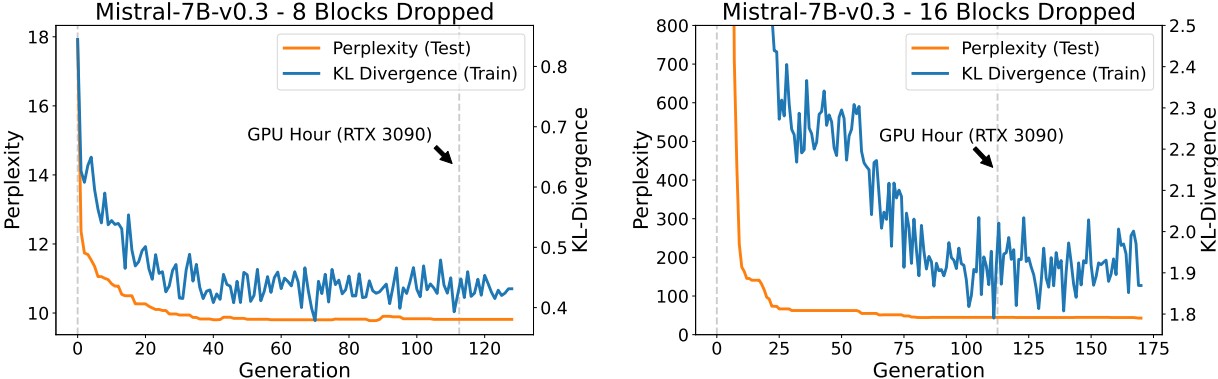

*Figure 8.* Convergence of EvoPress when removing 8 transformer blocks (left) and 16 transformer blocks (right) of Mistral-7B-v0.3.

## D.3. Locality of Dropped Blocks

Prior research indicates that deeper layers, aside from the final ones, are generally less effective (Gromov et al., 2024; Men et al., 2024). Figure 9 illustrates the optimal removal configurations identified by EvoPress. For comparison, Table 13 displays the removal order of prior scoring methods. While EvoPress indeed removes some deeper layers across all sparsities, we also observe that certain shallow layers appear to be less important. Notably, a "two hills" pattern emerges in many cases, where blocks before and after the midpoint are pruned, yet the central blocks remain intact. Meanwhile, the first two blocks are never pruned. However, in contrast to a heuristic proposed by Ma et al. (2023), we find that, in some instances, it is effective to prune the final block as well.

*Table 13.* First 16 blocks in removal order of ShortGPT, Weight Subcloning and Shortened Llama on different models.

| Model | Method | Removal Order (Left to Right) |
|---|---|---|
| Mistral-7B-v0.3 | ShortGPT | 26, 25, 24, 27, 23, 22, 28, 30, 21, 29, 20, 19, 13, 17, 18, 12 |
| | Weight Subcloning | 26, 25, 24, 27, 23, 28, 22, 30, 21, 29, 20, 19, 13, 17, 12, 18 |
| | Shortened Llama | 10, 12, 13, 11, 08, 09, 14, 15, 07, 06, 04, 27, 24, 16, 25, 05 |
| Llama-2-7B | ShortGPT | 27, 25, 26, 28, 29, 24, 23, 22, 21, 30, 20, 19, 18, 17, 15, 14 |
| | Weight Subcloning | 27, 25, 28, 29, 26, 24, 23, 22, 21, 19, 30, 20, 18, 17, 14, 15 |
| | Shortened Llama | 11, 12, 08, 09, 10, 06, 24, 25, 07, 14, 23, 13, 22, 21, 15, 27 |
| Llama-3-8B | ShortGPT | 25, 26, 27, 24, 28, 23, 22, 29, 20, 21, 19, 18, 30, 17, 16, 11 |
| | Weight Subcloning | 25, 27, 26, 24, 28, 23, 22, 29, 20, 21, 19, 18, 30, 17, 16, 11 |
| | Shortened Llama | 10, 08, 09, 11, 26, 25, 12, 22, 24, 23, 14, 13, 28, 06, 19, 21 |
| Llama-3.1-8B | ShortGPT | 25, 26, 24, 27, 23, 28, 22, 29, 20, 21, 19, 18, 17, 30, 16, 10 |
| | Weight Subcloning | 25, 27, 26, 24, 28, 23, 22, 29, 20, 21, 19, 18, 30, 17, 16, 10 |
| | Shortened Llama | 10, 09, 11, 08, 26, 25, 12, 24, 22, 23, 14, 28, 06, 13, 19, 21 |

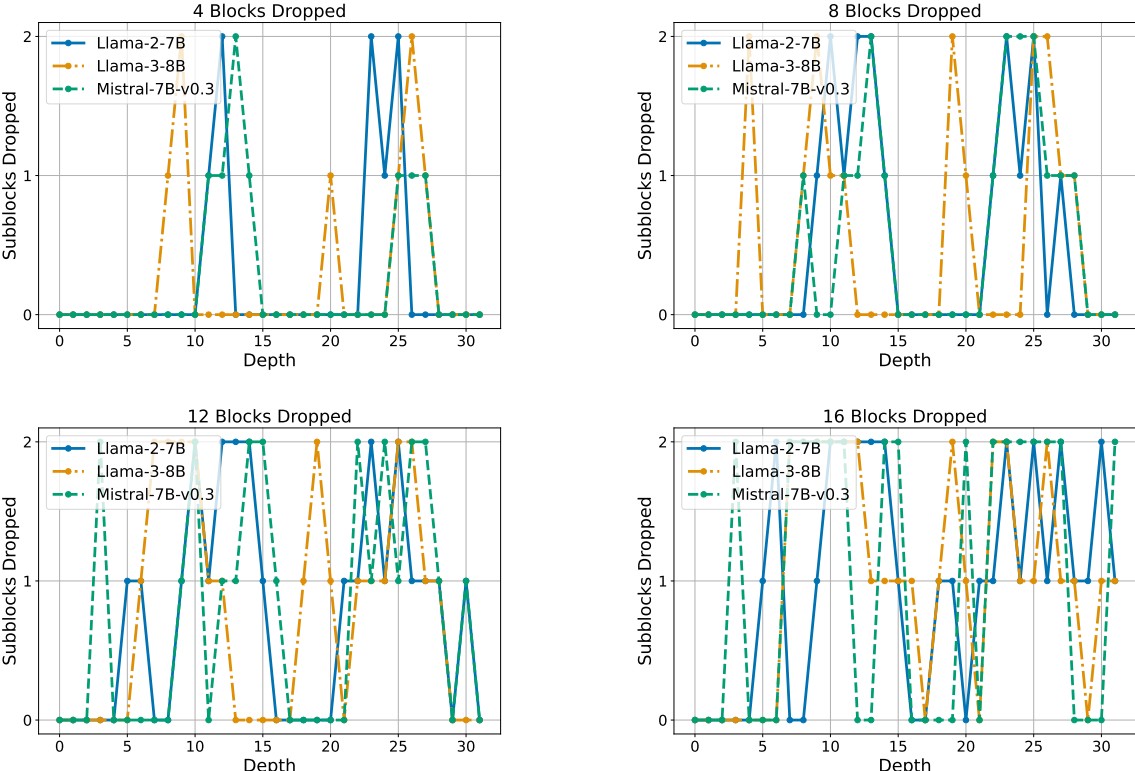

*Figure 9.* Optimal removal configurations identified by EvoPress for different models.

## D.4. Correlation of Scores with Perplexity

In this experiment, we first calculated the cosine similarity and squared error for each block by comparing activations before and after the block. Next, we randomly removed subsets of blocks (excluding the first and last two) and for each configuration, computed the average cosine similarity and squared error. The results are shown in Figure 11. Initially, the average squared error exhibited a negative correlation, as the $\ell_2$-norm of the activations increased with depth. This led to configurations with early blocks removed having small average error. To mitigate this, we normalized the activations prior to computing the squared error, which significantly improved the correlation, resulting in performance comparable to cosine similarity. However, as sparsity increased, the correlation degraded significantly for both methods, offering insight into why removal techniques based on scoring fail even at moderate levels of sparsity. Meanwhile, when removing only a small number of blocks, the average perplexity when removing each block separatly is a strong predictor of the performance after removing an entire set of blocks, as depicted in Figure 10. We conclude that error monotonicity holds at smaller compression levels, but decays rapidly at medium sparsity. The experiments were done using 131,072 tokens from the Fineweb-Edu calibration dataset.

*Figure 10.* Effect of removing random subsets of 2, 4, and 6 blocks for Llama-3-8B. The x-axis depicts the average perplexity when removing each block separately, while the y-axis depicts the perplexity after removing the entire set of blocks. Error monotonicity holds for smaller compression levels, but decays with increasing sparsity.

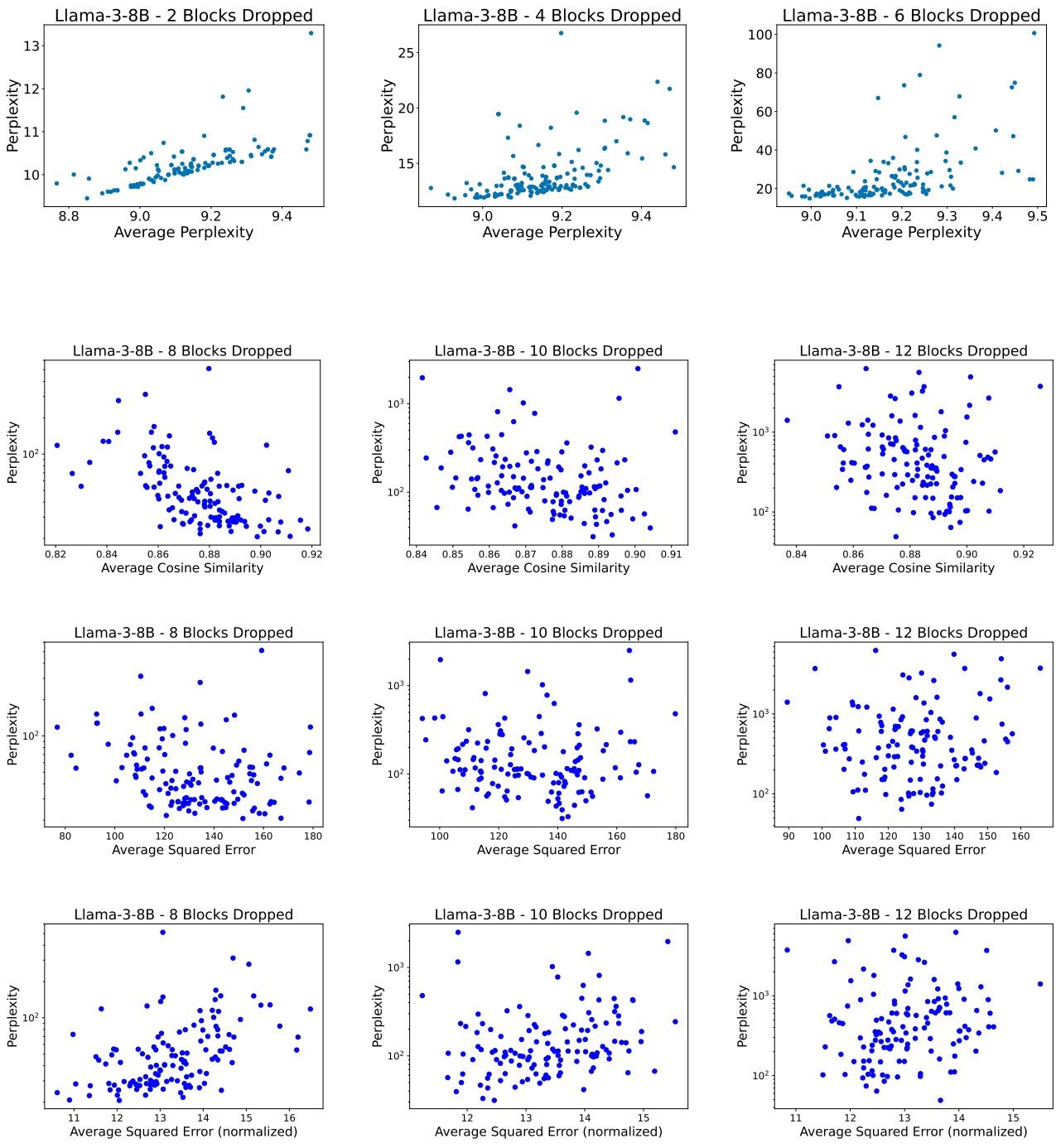

*Figure 11.* Effect of removing random subsets of 8, 10, and 12 blocks for Llama-3-8B. The x-axis depicts the average cosine similarity (first row), average squared error (second row), and the average squared error after normalization (third row) when removing each block separately, while the y-axis depicts the perplexity after removing the entire set of blocks. Error monotonicity breaks at medium sparsity rates, providing an explanation for the failure of scoring-based methods at these sparsity levels.

# E. Additional Unstructured Sparsity Results

## E.1. 50% and 60% Sparsity

In the main text, we focused on results at 70% sparsity, where the performance differences are more pronounced. However, since 50% and 60% sparsity levels are more practical and frequently referenced in the literature, we present the corresponding results in Tables 14 and 15. We have also included Llama-2-7B in these tables for legacy purposes. Even at these lower sparsity levels, EvoPress demonstrates significant improvements over uniform sparsity and consistently outperforms OWL.

*Table 14.* Performance of various sparsity profiles at 50% sparsity.

| Model | Method | Wiki2↓ | C4↓ | ArcC↑ | ArcE↑ | HS↑ | PiQA↑ | WG↑ | Avg↑ |
|---|---|---|---|---|---|---|---|---|---|
| Mistral-7B-v0.3 | Dense | 4.82 | 7.72 | 48.9 | 79.6 | 60.9 | 80.3 | 73.9 | 68.7 |
| | Uniform | 5.68 | 8.93 | 43.7 | 76.7 | 55.7 | 78.4 | 71.0 | 65.1 |
| | OWL | 5.69 | 8.94 | 43.9 | 76.9 | 55.4 | 78.5 | 70.3 | 65.0 |
| | EvoPress | **5.49** | **8.70** | 45.7 | 77.3 | 56.5 | 78.9 | 71.2 | **65.9** |
| Llama-2-7B | Dense | 5.12 | 6.93 | 43.4 | 76.3 | 57.1 | 78.1 | 69.0 | 64.8 |
| | Uniform | 6.40 | 8.87 | 41.3 | 73.4 | 52.8 | 75.7 | 68.8 | 62.4 |
| | OWL | 6.38 | 8.77 | 41.1 | 73.2 | 53.2 | 76.5 | 70.2 | 62.9 |
| | EvoPress | **6.22** | **8.52** | 41.5 | 74.2 | 54.0 | 76.7 | 69.6 | **63.2** |
| Llama-3-8B | Dense | 5.54 | 7.10 | 50.4 | 80.1 | 60.2 | 79.7 | 72.6 | 68.6 |
| | Uniform | 8.05 | 13.07 | 43.6 | 75.7 | 54.2 | 76.1 | 71.7 | 64.3 |
| | OWL | 8.13 | 13.12 | 43.8 | 75.8 | 54.0 | 75.7 | 72.2 | 64.3 |
| | EvoPress | **7.63** | **12.53** | 43.9 | 77.5 | 54.5 | 76.8 | 72.2 | **65.0** |
| Llama-3.1-8B | Dense | 5.61 | 8.90 | 51.2 | 81.4 | 60.0 | 80.1 | 73.9 | 69.3 |
| | Uniform | 8.06 | 13.03 | 44.5 | 76.7 | 54.0 | 76.7 | 71.5 | 64.7 |
| | OWL | 8.02 | 12.99 | 44.2 | 76.5 | 53.8 | 76.8 | 72.5 | 64.8 |
| | EvoPress | **7.51** | **12.31** | 46.6 | 77.7 | 54.9 | 77.6 | 71.7 | **65.7** |

*Table 15.* Performance of various sparsity profiles at 60% sparsity.

| Model | Method | Wiki2↓ | C4↓ | ArcC↑ | ArcE↑ | HS↑ | PiQA↑ | WG↑ | Avg↑ |
|---|---|---|---|---|---|---|---|---|---|
| Mistral-7B-v0.3 | Dense | 4.82 | 7.72 | 48.9 | 79.6 | 60.9 | 80.3 | 73.9 | 68.7 |
| | Uniform | 7.78 | 11.86 | 38.0 | 72.3 | 49.4 | 75.0 | 69.3 | 60.9 |
| | OWL | 7.50 | 11.34 | 38.5 | 71.9 | 49.6 | 75.1 | 70.2 | 61.1 |
| | EvoPress | **7.08** | **10.27** | 40.5 | 72.8 | 51.9 | 76.9 | 68.8 | **62.2** |
| Llama-2-7B | Dense | 5.12 | 6.93 | 43.4 | 76.3 | 57.1 | 78.1 | 69.0 | 64.8 |
| | Uniform | 9.3 | 12.37 | 35.8 | 69.5 | 45.9 | 72.4 | 65.9 | 57.9 |
| | OWL | 8.35 | 11.00 | 36.0 | 69.1 | 47.5 | 73.2 | 66.2 | 58.4 |
| | EvoPress | **8.21** | **10.34** | 37.1 | 70.6 | 49.3 | 74.4 | 67.6 | **59.8** |
| Llama-3-8B | Dense | 5.54 | 7.10 | 50.4 | 80.1 | 60.2 | 79.7 | 72.6 | 68.6 |
| | Uniform | 13.86 | 21.43 | 35.2 | 69.7 | 45.6 | 72.2 | 68.0 | 58.2 |
| | OWL | 12.37 | 18.53 | 38.0 | 70.3 | 47.7 | 72.1 | 68.5 | 59.3 |
| | EvoPress | **11.02** | **16.37** | 39.0 | 71.9 | 48.6 | 74.0 | 69.1 | **60.5** |
| Llama-3.1-8B | Dense | 5.61 | 8.90 | 51.2 | 81.4 | 60.0 | 80.1 | 73.9 | 69.3 |
| | Uniform | 13.43 | 21.46 | 36.4 | 69.7 | 46.2 | 72.3 | 67.7 | 58.5 |
| | OWL | 12.08 | 18.25 | 38.9 | 71.1 | 47.7 | 73.1 | 68.8 | 59.9 |
| | EvoPress | **10.58** | **15.96** | 40.0 | 72.5 | 49.0 | 74.6 | 69.5 | **61.1** |

## E.2. Sparsity Profiles

Below, we visualize sparsity profiles determined by EvoPress and baseline approaches. Notably, EvoPress prunes the initial blocks less aggressively than the middle and later blocks. Additionally, the `q_proj` projection attains higher sparsity levels, whereas the `v_proj` projection is pruned to significantly lower sparsity on average. Although Figure 12 may suggest that OWL and EvoPress produce similar sparsity profiles, this is misleading – OWL enforces uniform sparsity at block level, as their original per-layer approach underperformed (Yin et al., 2024).

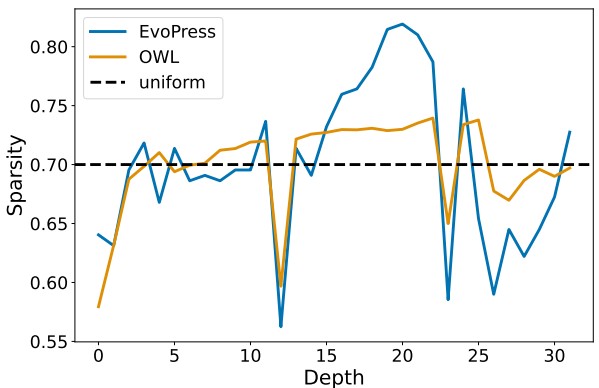

*Figure 12.* Comparison of different block-level sparsity profiles for Llama-3.1-8B at 70% sparsity.

*Figure 13.* Average sparsity per projection type for Llama-3.1-8B at 70% sparsity for EvoPress.

## F. Additional Quantization Results

### F.1. 2.25 Bit and 2.5 Bit

In addition to the 3 bit results presented in Section 4.3, we further evaluated EvoPress under extreme quantization conditions, specifically testing it at 2.25 bit and 2.5 bit levels. As a baseline, we generated 32 random configurations combining 2 bit and 3 bit layers and selected the best performing setup. The results, as shown in Table 16, demonstrate that EvoPress significantly outperforms this baseline, highlighting its ability to achieve extreme quantization levels.

*Table 16.* Performance of EvoPress on 2.25 bit and 2.5 bit quantization.

| Model | # Bits | Method | Wiki2↓ | C4↓ | ArcC↑ | ArcE↑ | HS↑ | PiQA↑ | WG↑ | Avg↑ |
|---|---|---|---|---|---|---|---|---|---|---|
| Mistral-7B-v0.3 | 2.25 | Best of 32 | 11.53 | 18.32 | 30.1 | 59.6 | 44.5 | 69.4 | 56.8 | 52.1 |
| | | EvoPress | **8.63** | **13.47** | 36.2 | 66.0 | 49.3 | 74.2 | 63.5 | **57.8** |
| | 2.5 | Best of 32 | 7.50 | 11.76 | 37.0 | 68.0 | 51.7 | 75.0 | 63.5 | 59.0 |
| | | EvoPress | **6.60** | **10.40** | 39.8 | 71.7 | 54.0 | 77.1 | 65.8 | **61.7** |
| Llama-2-7B | 2.25 | Best of 32 | 13.18 | 18.19 | 24.8 | 50.2 | 40.3 | 66.8 | 56.1 | 47.7 |
| | | EvoPress | **9.82** | **9.93** | 29.5 | 61.8 | 46.2 | 70.3 | 59.4 | **53.4** |
| | 2.5 | Best of 32 | 9.42 | 9.01 | 29.1 | 58.6 | 46.9 | 70.1 | 62.6 | 53.5 |
| | | EvoPress | **8.03** | **7.33** | 35.3 | 68.4 | 50.8 | 73.9 | 64.2 | **58.5** |
| Llama-3-8B | 2.25 | Best of 32 | 149.85 | 432.96 | 21.2 | 29.1 | 28.1 | 55.6 | 49.8 | 36.8 |
| | | EvoPress | **23.93** | **43.17** | 23.6 | 46.9 | 39.3 | 63.6 | 56.5 | **46.0** |
| | 2.5 | Best of 32 | 21.65 | 23.92 | 25.1 | 47.6 | 41.2 | 65.6 | 56.2 | 47.1 |
| | | EvoPress | **13.93** | **18.15** | 31.7 | 61.5 | 47.9 | 71.7 | 64.3 | **55.4** |
| Llama-3.1-8B | 2.25 | Best of 32 | 259.61 | 181.36 | 20.7 | 31.9 | 30.6 | 57.0 | 51.9 | 38.4 |
| | | EvoPress | **22.75** | **33.58** | 26.7 | 48.9 | 40.2 | 63.4 | 55.7 | **47.0** |
| | 2.5 | Best of 32 | 35.33 | 37.09 | 24.1 | 48.4 | 41.7 | 62.7 | 54.5 | 46.3 |
| | | EvoPress | **11.73** | **19.03** | 32.2 | 63.3 | 47.5 | 71.8 | 62.3 | **55.4** |
| Phi-3-Medium | 2.25 | Best of 32 | 14.20 | 18.19 | 28.9 | 46.8 | 40.0 | 61.8 | 53.1 | 46.1 |
| | | EvoPress | **10.48** | **14.60** | 36.2 | 62.0 | 46.6 | 66.2 | 55.6 | **53.3** |
| | 2.5 | Best of 32 | 8.26 | 12.65 | 40.5 | 69.3 | 50.3 | 70.9 | 61.9 | 58.6 |
| | | EvoPress | **7.12** | **11.23** | 44.1 | 75.9 | 54.1 | 73.5 | 64.6 | **62.4** |

## F.2. Practical Convergence

Similar to unstructured sparsity, EvoPress also demonstrates rapid convergence when applied to quantization. As shown in Figure 14, the majority of improvements occur within two GPU hours, with full convergence achieved after approximately eight GPU hours. If needed, this optimization time could be further shortened by tuning the hyperparameters, similarly to the "super-fast" version for unstructured sparsity discussed in Section 4.2. However, we observed that the convergence dynamics are less smooth compared to unstructured sparsity, likely due to the limited number of quantization levels available (practically only 2, 3, and 4 bit are used), which results in a less smooth fitness landscape.

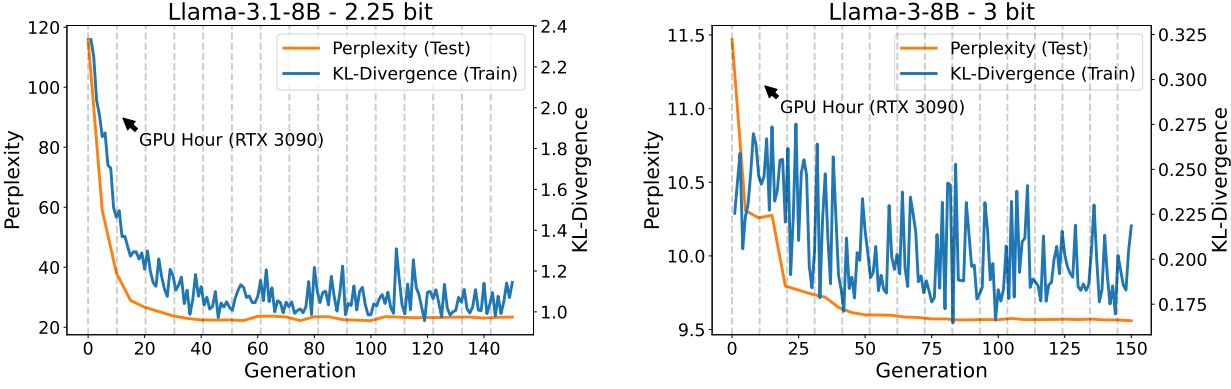

*Figure 14.* Convergence of EvoPress for 2.25 bit quantization on Llama-3.1-8B (left) and 3 bit quantization on Llama-3-8B (right).

## F.3. Quantization Profiles

In this section, we visualize a quantization profile determined by EvoPress. As shown, EvoPress maintains a relatively uniform quantization bitwidth allocation across the model. Interestingly, while the second and the two last blocks are less compressed, the first block undergoes significantly higher compression. This suggests that "saliency" does not directly

equate to block importance but rather depends on the specific compression method used. Additionally, similar to the profiles for unstructured sparsity, we observe a transfer of capacity to `v_proj`.

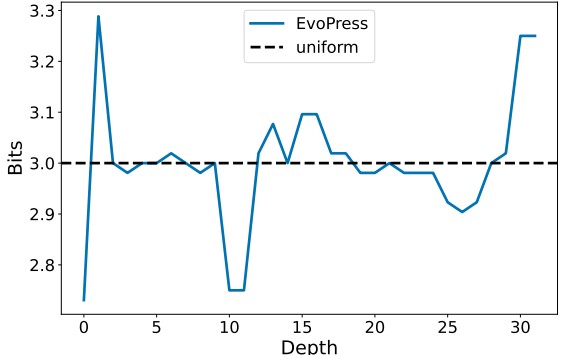

*Figure 15.* Block-level quantization profiles for Llama-3.1-8B at 3 bit compression on average.

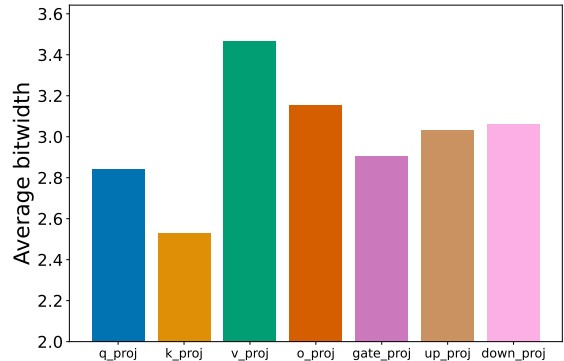

*Figure 16.* Average bitwidth per projection type for Llama-3.1-8B at 3 bit compression on average.

## G. Running Time Comparisons

In our main results, we did not include a stopping criteria into the optimization process, but instead continued the search well beyond convergence to observe the convergence behavior. Here, we include a simple stopping criteria to our search, which terminates the search whenever the perplexity on the training set has not improved in the last 20 generations. Based on this version we compare the runtime requirements of EvoPress with baseline methods. However, it should be noted that the amount of data used impacts both runtime and compression quality (with specific trade-offs being method-dependent), meaning one can generally trade-off both objectives, which makes runtime comparison difficult.

In Table 17 we compare EvoPress for block dropping to various baseline methods on Llama-2-7B, where we, additionally to the main results, also compare against the iterative search methods BlockPruner (Zhong et al., 2024) and SLEB (Song et al., 2024). As the results show, EvoPress achieves the best compressed models across all sparsity ratios, while being quicker than other search methods. Scoring-based methods are considerably faster than the search-based methods, but produce significantly worse models.

In Table 18 we provide a comparison of EvoPress with BESA (Xu et al., 2024), a method for non-uniform sparsity allocation. BESA performs gradient-descent based search for rowwise sparsity allocation using straight-through-estimation (Bengio et al., 2013), with uniform sparsity on a per-block level. Therefore, both methods are not fully comparable, as EvoPress searches for layerwise sparsities globally (across the entire model, not just blockwise), while keeping per-row sparsities uniform. We demonstrate that both methods are complementary by performing a two-stage search, first, for blockwise sparsities using EvoPress, then, for rowwise sparsities with fixed per-block sparsity using BESA. This merged version produces best results on 50% and 60% sparsity, while it performs slightly worse than pure EvoPress on 70% sparsity. For a fair comparison, we search on top of a Wanda (Sun et al., 2024) layer database for all methods. The runtime requirements for both BESA and EvoPress are similar, as there is an additional hyperparameter sweep for BESA, which is not provided in the author's implementation (authors provide tuned coefficients). Similarly, we have not included OWL in this comparison, as the authors have not provided instructions on how hyperparameters were obtained.

*Table 17.* Comparison of various depth pruning methods at different sparsities. We report the runtime on a single NVIDIA RTX 4090 GPU. Note that some methods perform forward passes on a strongly depth-pruned model, which explains the runtime difference for EvoPress on 50% and 12.5% sparsity.

| Sparsity | Method | Fineweb↓ | Wiki2↓ | C4↓ | Runtime | Fwd. Passes (Tokens) |
|----------|--------|----------|--------|-----|---------|----------------------|
| 0% | - | 6.40 | 5.12 | 6.99 | - | - |
| 12.5% | ShortGPT | 9.30 | 8.86 | 10.78 | 46s | 65K |
| 12.5% | Cosine Similarity (Window) | 8.51 | 7.53 | 9.82 | 46s | 65K |
| 12.5% | Weight Subcloning | 9.60 | 9.09 | 11.06 | 49s | 65K |
| 12.5% | ShortenedLlama | 8.57 | 7.68 | 10.44 | 8min | 2.1M |
| 12.5% | SLEB | 7.71 | 6.51 | 8.78 | 70min | 18.8M |
| 11.4% | BlockPruner | 7.88 | 7.88 | 8.86 | 100min | 26.6M |
| 12.5% | EvoPress | **7.61** | **6.41** | **8.67** | 58min | 12.4M |
| 25% | ShortGPT | 21.16 | 23.41 | 30.30 | 44s | 65K |
| 25% | Cosine Similarity (Window) | 17.37 | 16.60 | 21.04 | 46s | 65K |
| 25% | Weight Subcloning | 21.16 | 23.41 | 30.30 | 49s | 65K |
| 25% | ShortenedLlama | 11.81 | 13.86 | 14.08 | 8min | 2.1M |
| 25% | SLEB | 10.21 | **9.45** | 12.03 | 122min | 35.2M |
| 25% | BlockPruner | 11.31 | 12.49 | 13.32 | 152min | 42.8M |
| 25% | EvoPress | **10.06** | 10.31 | **11.99** | 44min | 10.0M |
| 37.5% | ShortGPT | 54.07 | 70.94 | 63.51 | 44s | 65K |
| 37.5% | Cosine Similarity (Window) | 151.10 | 192.07 | 212.60 | 45s | 65K |
| 37.5% | Weight Subcloning | 54.07 | 70.94 | 63.51 | 49s | 65K |
| 37.5% | ShortenedLlama | 20.37 | 35.37 | 26.07 | 8min | 2.1M |
| 37.5% | SLEB | 17.29 | 19.57 | 20.91 | 159min | 49.1M |
| 36.4% | BlockPruner | 19.41 | 28.75 | 22.44 | 185min | 54.8M |
| 37.5% | EvoPress | **16.09** | **17.44** | **19.87** | 53min | 19.7M |
| 50% | ShortGPT | 180.51 | 226.14 | 171.04 | 44s | 65K |
| 50% | Cosine Similarity (Window) | 3611.06 | 4570.15 | 2876.83 | 45s | 65K |
| 50% | Weight Subcloning | 180.51 | 226.14 | 141.04 | 49s | 65K |
| 50% | ShortenedLlama | 68.79 | 145.78 | 87.40 | 8min | 2.1M |
| 50% | SLEB | 51.30 | 117.21 | 58.56 | 184min | 60.1M |
| 48.9% | BlockPruner | 52.28 | 97.23 | 60.47 | 208min | 64.9M |
| 50% | EvoPress | **34.20** | **47.15** | **40.10** | 31min | 12.4M |

*Table 18.* Comparison of EvoPress with uniform pruning and BESA on Llama-2-7B. Both EvoPress as well as BESA (layerwise) operate on a layer database generated via Wanda. BESA (rowwise) additionally learns per-output-channel sparsities, which target a different dimension of the compression space than EvoPress's layerwise allocation. EvoPress+BESA applies BESA using the average per-block sparsities found from running EvoPress, showing that both methods can work complementary. Here, we use the lightweight version of EvoPress with early stopping.

| Sparsity | Method | Non-Uniformity | Fineweb↓ | Wiki2↓ | C4↓ | Runtime |
|---|---|---|---|---|---|---|
| 50.00 | Uniform | N/A | 7.60 | 6.41 | 8.97 | - |
| 49.79 | BESA | Layerwise | 7.55 | 6.32 | 8.93 | 16min* |
| 50.00 | EvoPress | Layerwise | 7.48 | 6.28 | 8.79 | 87min |
| 49.70 | BESA | Rowwise | 7.39 | 6.19 | 8.72 | 33min* |
| 49.72 | EvoPress+BESA | Rowwise | **7.37** | **6.16** | **8.71** | 120min |
| 60.00 | Uniform | N/A | 10.22 | 9.56 | 12.95 | - |
| 59.46 | BESA | Layerwise | 9.82 | 8.80 | 12.38 | 16min* |
| 58.91 | EvoPress | Layerwise | 8.75 | 8.03 | 11.94 | 102min |
| 59.99 | EvoPress | Layerwise | 9.01 | 8.66 | 10.96 | 94min |
| 58.90 | BESA | Rowwise | 9.15 | 8.11 | 11.88 | 33min* |
| 59.12 | EvoPress+BESA | Rowwise | **8.59** | **7.58** | **10.58** | 135min* |
| 70.00 | Uniform | N/A | 64.02 | 136.53 | 97.53 | - |
| 69.97 | BESA | Layerwise | 52.65 | 66.51 | 67.58 | 16min* |
| 68.52 | EvoPress | Layerwise | **13.53** | 15.61 | **16.90** | 111min |
| 69.99 | EvoPress | Layerwise | 15.55 | 24.04 | 19.54 | 120min |
| 68.51 | BESA | Rowwise | 16.53 | 17.79 | 24.09 | 33min* |
| 68.03 | EvoPress+BESA | Rowwise | 13.83 | **14.19** | 19.52 | 144min* |

˙ Sparsity values in red are more than 0.5% below the target. They are a result of BESA only enforcing a soft constraint on the effective sparsity.

* BESA runtimes do not include the hyperparameter sweep over '-d-coef'.

