# OpenReview forum: "EvoPress: Accurate Dynamic Model Compression via Evolutionary Search"
_ICML.cc/2025/Conference — ICML 2025 poster_

### Official Review · Reviewer_zC2u · 2025-02-24

**Overall Recommendation:** 4

**Summary:**

The paper proposes Evopress, a novel pruning approach for the dynamic compression of LLMs based on evolutionary computation. The authors identify a critical issue in current compression algorithm approaches: error monotonicity does not apply to LLM compression. Aiming to resolve such drawbacks, they propose a (1 + λ)-evolutionary algorithm for compressing LLMs, derived from the theory that the dynamic compression of LLMs has a linear fitness function. Hence, (1 + λ) sounds like an ad-hoc algorithm due to its hill-climbing properties, which are established to be optimal for linear fitness functions. The main novelty is that Evopress can be applied to different compression techniques: structured and unstructured pruning as well as quantization.
It also modifies the standard (1 + λ) by integrating level-switch mutation (to maintain a fixed sparsity ratio) and multi-step selection to overcome the high evaluation time required for the offspring.
The experimental results show how Evopress outperforms the selected baselines in terms of perplexity over the Language Modeling task and accuracy over Zero-Shot tasks.

## Update after rebuttal

The authors included in the rebuttal the required pruning runtime comparison. I stick to my final comment that the comparison should be done with the same calibration data size to have a fair comparison. All my other concerns were addressed. Hence, I changed my score from 2 to 4 and supported the acceptance of the paper. However, from my perspective, all the new results provided in the rebuttal should be included in the main paper. Also, a clear statement about the pruning runtime complexity should be highlighted in the main paper w.r.t. some more "pruning time" efficient baselines.

**Claims And Evidence:**

Partially.

The motivation for the linear landscape in dynamic compression is supported. Also, the numerical results over the tested task support the claims of the paper with respect to the selected baselines. A claim that, in my opinion, is not supported is "iteration efficient." While I agree with the authors that the multi-step selection improves the overall runtime of EvoPress, with respect to the baselines, the method is not efficient in terms of runtime and resources required to obtain the final sparse model. While this may not be considered a major problem, I personally don’t think that, in its current state, the paper properly highlights such limitations.
Another aspect that does not completely support the claims is the selection of baselines. While it is true that the paper tackles three different compression schemes, the baselines selected for each compression level do not provide a clear picture with respect to the recent literature for each compression scheme.

(See Relation To Broader Scientific Literature)

**Essential References Not Discussed:**

The paper does not discuss several works.

Regarding structured pruning, it does not mention or use [1,2] as baselines. For unstructured pruning, they apply Evopress to SparseGPT without mentioning [3,4]. On the quantization side, no recent weight-only quantization algorithms, such as [5,6,7], have been discussed or included in the baselines.

[1] Zhong, Longguang, et al. "Blockpruner: Fine-grained pruning for large language models." arXiv preprint arXiv:2406.10594 (2024). \
[2] Song, Jiwon, et al. "SLEB: Streamlining LLMs through Redundancy Verification and Elimination of Transformer Blocks." Forty-first International Conference on Machine Learning. \
[3] Zhang, Yingtao, et al. "Plug-and-play: An efficient post-training pruning method for large language models." The Twelfth International Conference on Learning Representations. 2024. \
[4]  Sun, Mingjie, et al. "A Simple and Effective Pruning Approach for Large Language Models." The Twelfth International Conference on Learning Representations. 2024 \
[5] Lin, Ji, et al. "Awq: Activation-aware weight quantization for on-device llm compression and acceleration." Proceedings of Machine Learning and Systems 6 (2024): 87-100. \
[6] Huang, Wei, et al. "BiLLM: Pushing the Limit of Post-Training Quantization for LLMs." Forty-first International Conference on Machine Learning \
[7]  Shao, Wenqi, et al. "OmniQuant: Omnidirectionally Calibrated Quantization for Large Language Models." The Twelfth International Conference on Learning Representations. \

**Experimental Designs Or Analyses:**

Yes. My main concern is the limited baselines for each compression scheme (see below)

**Methods And Evaluation Criteria:**

Yes, the selected models are in line with the latest public LLMs. The datasets and tasks are consistent with recent works on LLM compression.

**Other Comments Or Suggestions:**

I appreciated the idea of EvoPress and agree with the authors that it is the first framework that can be used across different compression schemes. I think the idea of using (1 + λ) as hill-climbing over a linear landscape, such as that of dynamic compression, is novel.
However, I would like to see a fairer comparison in terms of pruning runtime and resources required w.r.t. the baselines in order to increase my score. Even though the advantage of obtaining a sparse model compensates the time required to obtain it, I think it is fair to report the actual runtime (seconds) and resources (possibly FLOPs) required (as numerical values and not as vertical lines inside a plot) and especially w.r.t. the baselines. Also, provide a detailed explanation of why no other baselines are included, especially in the quantization experiments. Also, provide information about train and validation data; see the question below.

**Other Strengths And Weaknesses:**

## Strengths

- The investigation of the linear landscape for dynamic compression is novel and well-supported.
- Applying (1 + λ) for model compression in a linear landscape is novel and effective. The proposed level-switch and multi-step selection modifications allow (1 + λ) to be effective in compressing LLMs.
- Using KL-divergence instead of perplexity, as in [1,2], provides a smoother evaluation of the sparse model.
- The application of a single framework for different compression schemes is the main novelty and a key strength of the paper.
- The results across the selected baselines show that EvoPress achieves better performance.


## Major Weaknesses

- The main weakness, in my opinion, is the time required to obtain the sparse model. Figures 3 and 8 clearly show that EvoPress requires hours to complete its evolutionary process. In particular, Figure 3 (left) indicates that 13 hours are required. On the other hand, approaches like OWL only require a single forward pass of the calibration data through the model to obtain outlier values from which to compute block-wise sparsity. Similarly, structured pruning methods like ShortGPT and Sliding Window Cosine Similarity are time-efficient in computing block scores. While this may not be a reason to reject the paper, I believe a fairer comparison in terms of pruning runtime, along with clearer details on the time and resources needed to run EvoPress, is necessary.
- While I highly appreciate EvoPress for its ability to apply dynamic compression to different compression schemes, the evaluation of each scheme (in terms of baselines) is too limited.
- Regarding structured pruning, the paper claims, “All four previous methods rely on human-crafted,” however, Shortened Llama and [1,2] also rely on the perplexity metric to evaluate the sparse model, similar to the proposed KL-divergence approach.

**Questions For Authors:**

- In unstructured pruning, why apply EvoPress to SparseGPT instead of Wanda, which would have been faster, given that SparseGPT requires weight reconstruction differently for each level in the database?
- Is there a specific reason why there are no other baselines for quantization besides DP-pruning?
- Regarding the runtime plots, the authors state, “We observe convergence close to optimum in 5-6h” (Figure 3). However, when applying EvoPress, do you still run for all the generations specified in Table 8? I also have some doubts about the plot—does the train accuracy refer to the KL of the offspring at generation t (computed as an average across all individuals)? Is the test accuracy computed on the same calibration data used for KL-divergence? Please provide more information about it.
- Since (1 + λ) retains only one individual after evaluation, if the optimum is reached at generation t << num_generations, shouldn’t the evolutionary process stop? As you mention, EvoPress usually converges close to the optimum before reaching num_generations, why not implement a stopping condition?

**Relation To Broader Scientific Literature:**

The paper focuses on dynamic compression of both structured and unstructured pruning as well as quantization. It tackles the problem of monotonic error in current pruning approaches, and at the same time focuses on dynamic compression, which is a crucial topic at the moment in the pruning community.

**Theoretical Claims:**

I checked the Theorem 3.1 in the main text, not the full analysis in the Appendix.

---

> ### Author Rebuttal · Authors · 2025-04-01
>
> We thank the reviewer for a thorough and detailed view with a lot of insightful comments and suggestions. The concerns are addressed below:
>
> **Stopping Criteria**
>
> We ran the search for more generations than necessary to get insights about the convergence behavior. We agree that in practice one should determine the number of generations dynamically, and as suggested, we have implemented a stopping criteria: Every 20 generations the perplexity on the training set is computed, and if it hasn’t improved, the search terminates. We present timings and results of EvoPress including early stopping below.
>
> **Runtime vs. baselines**
>
> Determining meaningful runtime results for baselines is challenging due to several reasons:
>
> * For block dropping, 3 out of 4 baseline methods neither shared their code nor specified data usage, making runtime comparisons difficult. Generally, scoring methods are faster than EvoPress. We compare the runtime of EvoPress with baseline methods below.
> * For quantization and unstructured pruning, runtime depends on the specific method, while our approach works independently of the layerwise compression method. OWL, for instance, requires an extensive hyperparameter sweep (20 configurations) and a larger layer database than EvoPress, leading to similar runtime constraints in our SparseGPT setup, dominated by layer database generation.
> * Additionally, for pruning, the sweep in OWL is not provided by the authors, and it is not clear how the “best” model is chosen (in particular, over how much data). We decided (in their favor) to evaluate all 20 models on the entire evaluation data, and choose the model with lowest perplexities.
>
> We will make this more transparent in the revised version. To address the reviewer’s concern, we have included runtimes under common parameter values below.
>
> **Comparison with baselines**
>
> *Block dropping*
>
> We provide comparisons with SLEB and BlockPruner in [Table](https://anonymous.4open.science/r/EvoPress-Data-DF25/tables/Comparison_depth_pruning.pdf). We also included the runtimes of other baseline methods, where we used 64K calibration tokens of Fineweb Edu consistently. One can observe that EvoPress outperforms these additional baselines, especially at higher compression, while being faster. We report the results of EvoPress with early stopping as outlined above.
>
> *Unstructured sparsity*
>
> Plug and Play and Wanda are relevant to unstructured pruning, and we'll include them in the new revision. Our work primarily focuses on finding the optimal sparsification profile on top of a layerwise compression method. We adopted SparseGPT due to its popularity and good performance. We demonstrate that EvoPress also functions effectively on top of a layer database generated using Wanda in [Table](https://anonymous.4open.science/r/EvoPress-Data-DF25/tables/Besa_wanda_table.pdf), where we also report runtime measurements. As requested by Reviewer **eQfQ**, the table contains results for BESA.
>
> *Quantization*
>
> AWQ, BiLLM, and OmniQuant will be discussed in our updated version, acknowledging their relevance. To test EvoPress's compatibility with various quantization methods, we applied it to a database generated by AWQ. Due to AWQ's constraints, the search space requires the same bitwidth within transformer block. Despite this, EvoPress achieved an improved compression profile (see [Table](https://anonymous.4open.science/r/EvoPress-Data-DF25/tables/EvoPress-AWQ.pdf)).
>
> **Human-Crafted Scoring**
>
> We wanted to emphasize the fact that the baseline methods do not evaluate the fully pruned model, instead, they make strong assumptions on how the fully pruned model performs based on another metric, which is inherent to scoring methods. In contrast, EvoPress (and [1,2] in a more restricted setting) evaluate the fully pruned models directly. We will rephrase this in the revised version.
>
> **Questions**
>
> * *Use of SparseGPT*. Regarding SparseGPT, we used it for its reliable performance at high compression, superior to Wanda at 60%-70% sparsity, which are our targets. Wanda, however, allows re-use of weight configurations across sparsities with separate masks.
> * *Mixed-quantization baselines*. At the time of submission, we were not aware of any baselines that address the problem of mixed-precision quantization in a comparable setting. Reviewer **eQfQ** suggested the Slim-LLM paper, which we compared but found it complementary due to its channel-wise approach. OWL's layerwise non-uniform quantization performed worse than uniform quantization, so we did not include it.
> * *Convergence Plots of EvoPress*. The train accuracy refers to the KL-Divergence of the survivor of the selection process, i.e. the parent of the subsequent iteration. It is measured on a random subset of the full training data (Fineweb-Edu), which is why the line is noisy. The test accuracy is measured on the full set of hold-out Fineweb-Edu data and does not impact the search process. We will clarify this in the revision.

---

> > ### Comment · Reviewer_zC2u · 2025-04-01
> >
> > Dear Authors,
> > Your rebuttal addressed all of my concerns. Just a couple of minor comments:
> >
> > - Comparison_depth_pruning.pdf: It would be better to evaluate performance and runtime using the same number of tokens for the forward pass. This would make the comparison clearer, but otherwise, it could be noisy.
> > - Nice to hear that the stopping condition can improve the runtime performance of your algorithm. I suggest including it in the main paper and the official codebase.
> > I suggest the authors include all the additional results and discussion used during the rebuttal in the main paper.
> >
> > I will change my score accordingly and support the acceptance of this work.

---

### Official Review · Reviewer_vouD · 2025-03-12

**Overall Recommendation:** 3

**Summary:**

This paper proposes EvoPress, a general framework for LLM compression. The authors observe that error monotonicity does not hold for LLM, and proposes an evolutionary search approach to improve performance. Experimental results demonstrate that on three compression approaches, depth pruning, unstructured sparsity, and quantization, the proposed methods achieve state-of-the-art performance for dynamic compression.

**Claims And Evidence:**

Yes, the evidence provided by the authors supports the claims.

**Essential References Not Discussed:**

I'm not familiar with this area, so I'm not sure.

**Experimental Designs Or Analyses:**

I think the experimental designs in this paper are reasonable.

**Methods And Evaluation Criteria:**

Yes, I think it makes sense.

**Other Comments Or Suggestions:**

N/A

**Other Strengths And Weaknesses:**

Strengths:

The proposed method is applicable to depth pruning, unstructured sparsity, and quantization while achieving strong performance.

Weaknesses:
1. The theoretical proof does not fully explain its effectiveness in nonlinear LLM compression. It is unclear whether EvoPress maintains optimization capability in nonlinear regions.
2. The proposed strategy may get trapped in local optima in high-dimensional and multimodal compressed spaces. Its performance in highly constrained or multi-objective settings remains uncertain.

**Questions For Authors:**

See the weakness.

**Relation To Broader Scientific Literature:**

EvoPress greatly expands the literature by combining evolutionary optimization with dynamic LLM compression, challenging long-held heuristic assumptions, and providing a flexible framework.

**Theoretical Claims:**

The mathematical symbols, variables, and equations are generally well defined and mathematically correct.

---

> ### Author Rebuttal · Authors · 2025-04-01
>
> We thank the reviewer for their comments. Below, we address the two weaknesses:
>
> > The theoretical proof does not fully explain its effectiveness in nonlinear LLM compression. It is unclear whether EvoPress maintains optimization capability in nonlinear regions.
>
> Fundamentally this is true – we cannot expect the linearity assumption to fully hold in practice. Notably, assuming linearity is standard in prior work, such as OWL, ShortGPT, and DP-based methods such as SPDY. In contrast, while our theoretical results rely on this assumption, EvoPress can optimize a much larger class of fitness environments. In this sense, EvoPress operates under significantly weaker assumptions than prior approaches. That said, our main contribution is empirical, with EvoPress showing strong performance across various settings.
>
> Additionally, we conducted experiments to verify the capabilities of EvoPress:
> * In Figure 1, we consider the problem of removing twelve transformer blocks of Llama-3-8B, with the additional constraint that only pairs of consecutive blocks can be removed. We brute forced the entire search space (16 choose 6 = 8008 configurations) using significant compute in order to identify the global optima. Then, we ran our evolutionary approach on this problem, and found that it detects the global optima within 6 generations.
> * We performed a robustness analysis for pruning Llama-3-8B to 70% sparsity. We found that over 16 independent runs the resulting perplexity on hold-out data has very little variance (Figure 7), and the final configurations show high similarity (Figure 6). This indicates that local optima do not pose a problem for the search process.
>
> > The proposed strategy may get trapped in local optima in high-dimensional and multimodal compressed spaces. Its performance in highly constrained or multi-objective settings remains uncertain.
>
> We would like to point out that the search space is relatively high-dimensional - the dimensionality of space is several dozens for the case of block dropping and several hundreds for quantization and unstructured sparsity. For a high enough compression ratio the loss landscape is nonlinear (as the example in Table 1 shows), yet the evolutionary search procedure successfully converges to a good optimum. EvoPress is compatible with multiple compression techniques simultaneously. Specifically, we conducted an experiment with joint depth pruning (with 25% blocks dropped) and quantization (with 4 bits on average given 2, 3, 4, 5, 6 bit width options) (see details in response to reviewer **Q9Fo**).

---

### Official Review · Reviewer_eQfQ · 2025-03-13

**Overall Recommendation:** 3

**Summary:**

This paper aimed at LLM compression and motivated by the observation that depth pruning LLM further may improve the performance. Evolutionary search algorithm is applied to search pruned model with compressed size and performance constraint. It also applied to layer/block-wise non-uniform unstructured pruning and quantization. The results show the improvement for these three compression methods.

**Claims And Evidence:**

The paper claims SOTA performance on depth pruning, unstructured pruning and quantization but there are some related works not compared in this paper.

**Essential References Not Discussed:**

- For depth pruning, SLEB[1] and Slicegpt[2] should be compared.
- For unstructured pruning, BESA[3] should be compared.
- For quantization, other mixed-precision quantization works should be discussed and compared, such as SliM-LLM[4], CWMQ[5].

[1] SLEB: streamlining llms through redundancy verification and elimination of transformer blocks.

[2] Slicegpt: Compress large language models by deleting rows and columns.

[3] BESA: Pruning Large Language Models with Blockwise Parameter-Efficient Sparsity Allocation

[4] SliM-LLM: Salience-Driven Mixed-Precision Quantization for Large Language Models

[5] Channel-Wise Mixed-Precision Quantization for Large Language Models

**Experimental Designs Or Analyses:**

Only application 2 (unstructured sparsity) presents the running time for the proposed method and running time for depth pruning and quantization are not clear.
In Figure3, the “super-fast” version seems not reach out the same model performance compared with the normal version.

**Methods And Evaluation Criteria:**

Yes.

**Other Comments Or Suggestions:**

N/A

**Other Strengths And Weaknesses:**

N/A

**Questions For Authors:**

N/A

**Relation To Broader Scientific Literature:**

Layer doping, pruning and quantization are all efficient methods for LLM memory problem as related works referenced in the paper.

**Theoretical Claims:**

Yes. The evolutionary search algorithm is checked.

---

> ### Author Rebuttal · Authors · 2025-04-01
>
> We thank the reviewer for the feedback. The concerns are addressed below:
>
> **Experimental design and analyzes**
>
> *Runtime of the method*
>
> Similar as for unstructured pruning in the main text, we have indicated the runtime of EvoPress for block dropping and quantization within convergence plots (see Appendix Figure 8 for Block Dropping, and Appendix Figure 14 for Quantization). We provide runtime measurements in the additional baseline comparisons further down.
>
> You correctly observed that the super-fast version depicted in Figure 3 does not fully reach the model performance of the more heavy-weight search. However, the hit in performance is rather small, and we wanted to demonstrate that it can be obtained for a fraction of the search time. This also suggests a practical strategy: one could first run the faster search to quickly reach good performance, and only then switch to the more heavy-weight search for best results.
>
> **Comparison with Baselines**
>
> We acknowledge your point; below, we provide additional comparisons with existing methods.
>
> 1. Depth pruning
>
> We acknowledge that SLEB is a relevant baseline and will include it as related work in the revision. Below, we have included a comparison with EvoPress on Llama2-7B, where we also included BlockPruner as a baseline, following the suggestion of Reviewer **zC2u**. Both SLEB and BlockPruner iteratively remove blocks and can thus be interpreted as highly restricted search procedures. In such methods, once a block is removed, it cannot be recovered in later iterations, which explains the worse performance compared to EvoPress. We report the runtime of EvoPress when using a simple stopping criteria, that terminates the search whenever the perplexity on the training data has not improved within the last 20 generations. The comparison of different depth pruning methods is detailed in [Table](https://anonymous.4open.science/r/EvoPress-Data-DF25/tables/Comparison_depth_pruning.pdf).
>
> 2. Unstructured sparsity
>
> Indeed, BESA is a relevant work and will be added to the related work section in the revised edition. For a fair comparison, we ran the fast version of EvoPress on a Wanda layer database (thus, the results are slightly worse than presented in the paper, where we used SparseGPT). Still, EvoPress finds notably improved layerwise sparsity allocations.
>
> For completeness, we also included the row-wise version of BESA, which adapts the Wanda Mask within each layer. We show that this is complementary to EvoPress by using the average per-block sparsity found by EvoPress as the per-block sparsity for BESA (EvoPress+BESA), which mostly yields better perplexities than BESA and EvoPress individually. However, this is still very much experimental, and there might be better methods to merge both approaches, like producing a layer database using BESA with row-wise non-uniform sparsity allocation, and then searching this database with EvoPress.
>
> The experimental results with BESA in the original setup as well EvoPress applied on top of BESA and Wanda are provided in [Table](https://anonymous.4open.science/r/EvoPress-Data-DF25/tables/Besa_wanda_table.pdf).
>
> 3. Quantization
>
> Slim-LLM and CWMQ perform mixed precision quantization on a per-channel level compared to the per-layer precision allocation adopted in our method. Therefore, we could adopt Slim-LLM or CWMQ as an alternative to GPTQ, and run EvoPress on top of this method. A direct comparison between EvoPress and these two works is not possible. However, we believe that EvoPress would synergize very well with such intra-layer mixed precision quantization methods, as it allows to produce layers with non-integer bitwidth, which is beneficial for the search procedure. According to your suggestion, we will reference these two works in the updated revision.
>
> *Slim-LLM*
>
> We conducted Slim-LLM experiments using the open-sourced implementation on GitHub. We considered two setups:
> - Original calibration data (Wikitext2, 128k samples of length 2048)
> - Our calibration data (FinewebEdu, 8M tokens)
>
> EvoPress shows significant improvement over SlimLLM in the paper's original setup in terms of 0-shot and MMLU evaluations. When comparing methods with the same calibration data, both methods show similar performance. However, the runtime of Slim-LLM increases dramatically with more data (~50 hours on single RTX3090 GPU compared to ~10 hours for EvoPress). That said, EvoPress and SlimLLM are complementary and hence, running EvoPress on top of SlimLLM will lead to further gains. We present results in [Table](https://anonymous.4open.science/r/EvoPress-Data-DF25/tables/Slim-LLM_vs_EvoPress.pdf).
>
> *CWMQ*
>
> The algorithm implementation of CMWQ is not open-sourced, making it difficult to provide a comparison in a comparable setup.

---

### Official Review · Reviewer_Q9Fo · 2025-03-14

**Overall Recommendation:** 4

**Summary:**

The paper introduces EvoPress for dynamic LLM compression, which optimizes compression levels across different model components to minimize accuracy loss while satisfying a global compression threshold. By formulating dynamic compression as an optimization problem, EvoPress efficiently determines optimal compression profiles. Experiments on multiple models demonstrate SOTA results for LLM compression.

**Claims And Evidence:**

Yes. Experiments demonstrate that EvoPress has SOTA performance for dynamic compression on various models.

**Essential References Not Discussed:**

Most related works are discussed in paper.

**Experimental Designs Or Analyses:**

The overall experiments are valid. Various models and different compression methods are tested.

**Methods And Evaluation Criteria:**

The proposed methods and evaluation make sense. Here are some questions for the proposed method:

(1) To quantify model degradation, KL divergence was used in Section 3. KL divergence is popular to do it, but there are other ways, for example max absolute value, or we can even use a small calibration dataset to measure perplexity, etc. It’ll be great if authors could explain this design choice.

(2) The paper has a high level problem definition. However, the proposed approach was only applied to single compression approach one by one (pruning or sparsity or quantization). It will be great if authors could show or explain the results of applying EvoPress on multiple compression approaches simultaneously.

**Other Comments Or Suggestions:**

Please see the comments above.

**Other Strengths And Weaknesses:**

Please see the comments above.

**Questions For Authors:**

Please see the comments above.

**Relation To Broader Scientific Literature:**

LLM model compression under a threshold is very important. This paper proposed an efficient method to solve this problem, and has the potential to inspire future works on dynamic model compression.

**Theoretical Claims:**

Briefly read the proofs in Section 3 and appendix.

---

> ### Author Rebuttal · Authors · 2025-04-01
>
> We thank the reviewer for the feedback. The questions are addressed below:
>
> > To quantify model degradation, KL divergence was used in Section 3. KL divergence is popular to do it, but there are other ways, for example max absolute value, or we can even use a small calibration dataset to measure perplexity, etc. It’ll be great if authors could explain this design choice.
>
> In our work we considered both KL-Divergence and Perplexity on the calibration set as fitness functions in the evolutionary search. We provide an ablation on these choices in Appendix B.3. Both choices show similar performance, but we decided to stick to KL-Divergence as it performed slightly better. The comparison between these two options is provided in **Table 1**.
>
> **Table 1. Comparison between KL-Divergence and Perplexity as the fitness function**
>
> | Model      	| # Bits | Method    	| Wiki2↓ | C4↓   | FW↓   |
> |----------------|--------|---------------|--------|-------|-------|
> | Llama-3-8B 	| 3  	| Uniform   	| 12.19  | 15.76 | 11.47 |
> |            	|    	| EvoPress (PPL)| 8.17   | 12.15 | 9.64  |
> |            	|    	| EvoPress (KL) | **7.49**   | **12.03** | **9.56**
> | Llama-2-7B 	| 3  	| Uniform   	| 6.16   | 7.96  | 6.86  |
> |            	|    	| EvoPress (PPL)| 5.74   | 7.90  | 6.79  |
> |            	|    	| EvoPress (KL) | **5.70**   | **7.87**  | **6.76**  |
> | Mistral-7B-v0.3| 3  	| Uniform   	| 5.54   | 8.57  | 6.96  |
> |            	|    	| EvoPress (PPL)| 5.23   | 8.45  | 6.87  |
> |            	|    	| EvoPress (KL) | **5.21**   | **8.42**  | **6.86**  |
>
>
> We believe this is because KL-Divergence measures relative degradation to the base model, which is more informative than a pure next token loss. This is especially important when using as little calibration data as we do in our multistep selection process. We decided to adopt KL-Divergence to quantify the distance between the two predictive distributions, as it is a well-established metric, with a grounding in information theory.
>
> > The paper has a high level problem definition. However, the proposed approach was only applied to single compression approach one by one (pruning or sparsity or quantization). It will be great if authors could show or explain the results of applying EvoPress on multiple compression approaches simultaneously.
>
> EvoPress is compatible with multiple compression approaches simultaneously. We conducted an experiment with joint depth pruning (with 25% blocks dropped) and quantization (with 4 bits on average given 2, 3, 4, 5, 6 bit width options). Specifically, on each step of the evolutionary algorithm we perform an alternating optimization between block dropping and quantization. Firstly, we select the optimal depth pruning and configuration and then exchange quantization bit-width between “alive” blocks. Our experimental results (see [Wikitext-2 perplexity](https://anonymous.4open.science/r/EvoPress-Data-DF25/figures/multimodal_search_wikitext2.pdf) and [C4 perplexity](https://anonymous.4open.science/r/EvoPress-Data-DF25/figures/multimodal_search_c4.pdf)) suggest that EvoPress manages to find a better solution given some starting point and exhibits relatively stable convergence.

---

### Decision · Program_Chairs · 2025-05-01

**Decision:**

Accept (poster)

**Comment:**

This paper proposed a method called EvoPress, an evolutionary search-based framework for dynamic LLM compression, capable of optimizing depth pruning, unstructured sparsity, and quantization under global accuracy constraints. Experimental results looked strong. All reviewers agreed that this is a good contribution to the community and authors have addressed the concerns during the rebuttal phase.